# MACHINE UNLEARNING FOR STREAMING FORGETTING

## ABSTRACT

Machine unlearning aims to remove knowledge derived from the specific training data that are requested to be forgotten in a well-trained model while preserving the knowledge learned from the remaining training data. Currently, machine unlearning methods typically handle all forgetting data in a single batch, removing the corresponding knowledge all at once upon request. However, in practical scenarios, requests for data removal often arise in a streaming manner rather than in a single batch, leading to reduced efficiency and effectiveness in existing methods. Such challenges of streaming forgetting have not been the focus of much research. In this paper, to address the challenges of performance maintenance, efficiency, and data access brought about by streaming unlearning requests, we introduce an online unlearning paradigm, formalizing the unlearning as a distribution shift problem. We then estimate the altered distribution and propose a novel online unlearning algorithm to achieve efficient streaming forgetting without requiring access to the original training data. Theoretical analyses confirm an $O(V_T\sqrt{T} + \Delta_T)$ error bound on the streaming unlearning regret, where $V_T$ represents the cumulative total variation in the optimal solution over $T$ learning rounds and $\Delta_T$ represents the cumulative total divergence between remaining and forgetting data distributions. This theoretical guarantee is achieved under mild conditions without the strong restriction of convex loss function. Experiments across various models and datasets validate the performance of our proposed method.

## 1 INTRODUCTION

Machine unlearning aims at safeguarding the privacy rights of individuals concerning sensitive and private data (Voigt & Von dem Bussche, 2017; Bourtoule et al., 2021; de la Torre, 2018). The objective of machine unlearning is to remove information associated with a selected group of data, referred to as *forgetting data*, from a well-trained model while retaining the knowledge encapsulated in the *remaining data*. (Bourtoule et al., 2021). Presently, research in this field has made some progress in designing effective unlearning algorithms. Current unlearning methodologies typically consider the forgetting data as a single batch and approach unlearning as a singular adjustment process. This process removes all information from the forgetting data at once, then uses the remaining data to repair and update the model, preserving its functionality (Kurmanji et al., 2023; Chen et al., 2023).

In practical scenarios, data removal requests from sensitive information owners are usually made incrementally, rather than in predetermined batches. For example, social media users might request the deletion of recommendations learned from their personal browsing history at any time, resulting in a continuous stream of individual requests. This streaming nature means that requests are submitted immediately as users identify their needs and often arise in a streaming manner rather than being grouped and processed together. To address the streaming forgetting problem, where removal requests occur incrementally or in streams, existing batch unlearning approaches often handle each request from scratch, one by one. This method faces several issues, especially when requests are frequent.

The first issue is the accumulated performance drop. Although the performance degradation usually happens in machine unlearning, previous methods have tried to reduce the degradation in the batch unlearning (Bourtoule et al., 2021; Chundawat et al., 2023a; Chen et al., 2023; Graves et al., 2021; Thudi et al., 2022; Shen et al., 2024b). However, this degradation can accumulate across multiple rounds, leading to a significant decline in overall model performance over time. The second issue is efficiency, which stems from the repeated performance repairs on the remaining data (Bourtoule et al., 2021; Chundawat et al., 2023a; Chen et al., 2023; Shen et al., 2024b). Due to the high

overlap of remaining data across different rounds, there are significant time and computational costs associated with reprocessing the same data, making the unlearning process inefficient. In addition, frequent access to the remaining data can be problematic due to data regularization policies Voigt & Von dem Bussche (2017); de la Torre (2018). In many cases, parts of the training data may no longer be accessible or may be subject to strict access controls (Sekhari et al., 2021; Chen et al., 2023; Chundawat et al., 2023b), hindering the necessary updates and repairs to the model. These limitations emphasize the need for a novel streaming forgetting method that can handle long sequences of data removal requests without performance degradation (Shen et al., 2024a; Gupta et al., 2021). It should optimize time and memory consumption (Nguyen et al., 2022; Tarun et al., 2023) with minimal reliance on the training data. Although some prior works have explored stream forgetting in the context of meta-learning tasks (Chen et al., 2022) and ensemble models (Liu et al., 2022), these approaches predominantly focus on model forgetting rather than data forgetting. Only two studies have directly tackled the problem of stream data forgetting (Zhao et al., 2024; Li et al., 2021), both of which impose specific constraints on model structures to realize unlearning (Zhao et al., 2024; Li et al., 2021). As a result, the problem of streaming unlearning for data instances from general well-trained models remains an open research challenge.

In this paper, we propose a novel streaming unlearning method that addresses the accumulated drops in both effectiveness and efficiency while reducing the need for frequent access to training data. To estimate the unlearning risk without training data, we formalize unlearning as a distribution shift problem. The shifted distribution caused by removing forgetting data serves as prior knowledge to make the unlearning process more efficient and accurate. To incrementally update the model towards unlearning, we propose a risk estimator to achieve the optimal model in each streaming round and propose the corresponding streaming unlearning approach – SAFE (**S**tream-**A**ware **F**org**e**tting). Our approach departs from traditional batch unlearning by incorporating dynamic regret risk and reducing reliance on original training data. Furthermore, our theoretical analysis guarantees the effectiveness of SAFE by providing the upper bound on the unlearning regret risk of $O(V_T\sqrt{T} + \Delta_T)$, where $V_T$ represents the cumulative total variation in the optimal solution over $T$ learning rounds, and $\Delta_T = \sum_{t=1}^{T} \mathrm{div}(D_t, F_t)$ represents the cumulative divergence between the remaining and forgetting data distributions. This result holds without assuming the convexity of the loss function. To evaluate the practical performance of SAFE, we conduct empirical experiments on both basic machine learning models and deep neural networks across various datasets.

The contributions of this paper can be summarised as follows:

- We introduce the online unlearning paradigm and the SAFE algorithm to address the streaming unlearning problem. SAFE maintains high predictive performance on the remaining data while ensuring high unlearning efficiency. Notably, it does not require repeated access to the original training data during unlearning.
- We are the first to provide an $O(V_T\sqrt{T} + \Delta_T)$ upper bound on the unlearning regret risk of the proposed algorithms through theoretical analysis, showing that the unlearning performances are proportion to the distribution divergence of remaining and forgetting data.
- Through empirical evaluations across multiple datasets and models, we demonstrate that SAFE achieves higher or comparable performance more efficiently than other state-of-the-art batch unlearning methods, especially on neural network-based models.

## 2 PRELIMINARIES AND BACKGROUND

### 2.1 PRELIMINARIES

**Batch Machine Unlearning Objective** We begin by introducing traditional batch unlearning. We consider a supervised learning task, where an initial model $f(\cdot; w)$ is trained on a dataset $D$ with the loss function $\ell(f(\cdot; w), \cdot)$. When a subset $F \subset D$ is selected as the forgetting data, an unlearning algorithm is applied to remove the knowledge associated with $F$ from the model $f$. The updated parameters $w$ of the model after unlearning are obtained by solving the following optimization problem:

$$\min_{w} \left( \mathcal{L}(D - F, w) + \lambda \mathcal{R}(F, w) \right), \tag{1}$$

where $\mathcal{R}(F, w)$ is a regularizer, and $\lambda$ is a trade-off weight hyperparameter for the regularizer. While minimizing the training loss $\mathcal{L}(D - F, w)$ on the remaining data $D - F$ ensures the unlearned classifier's performance on the retaining data, the regularizer is used to control the performance of forgetting data. Specifically, in this paper, we define it as $\mathcal{R}(F, w) = 1/|F| \sum_{(\mathbf{x}, y) \in F} d_{\mathrm{KL}}(f(\mathbf{x}; w), f(\mathbf{x}; w^*))$ where $w^*$ is the optimal model parameters for minimizing $\mathcal{L}(D - F, w)$ and $d_{\mathrm{KL}}$ defines the Kullback-Leibler (KL) Divergence. It constraints that the model after unlearning should achieve similar performance as $f(\cdot; w^*)$. When $\lambda$ is zero, the unlearning objective equals the classical retraining-based one used in (Zhang et al., 2024; Thudi et al., 2022); when $\lambda$ is set as other positive value, the performance on forgetting data is specially incorporated into consideration, as shown in (Fan et al., 2024; Kurmanji et al., 2023).

**Online Learning with Dynamic Regret**   An online learning problem shares the same streaming data nature as streaming unlearning but differs in how the provided data is used. In online learning, the model is sequentially updated by incorporating new data. Specifically, in online learning, the training set is augmented in the $t$-th round, denoted as $D_t$. The goal is to minimize the dynamic regret, which is

$$L\text{-}Regret_T = \sum_{t=1}^{T} (\mathcal{L}(D_t, w_t) - \mathcal{L}(D_t, w_t^\dagger)), \tag{2}$$

where $w_t^\dagger$ denotes the optimal parameters in the $t$-th round of online learning (Hoi et al., 2021; Zinkevich, 2003a; Besbes et al., 2015). Here, the dynamic regret represents the cumulative risk between the online models and the optimal models in each round. Such principle of regrets, which defines the distance between the performances of the updated model and optimal model can also be adapted into unlearning as an objective. We will give details on how to design a regression objective for online unlearning and its difference from the regret in online learning in the following subsection

## 2.2 Online Unlearning Paradigm

In the streaming unlearning problem, the original training dataset $D_0$ consists of data points $(\mathbf{x}, y) \in D_0$ is used to train an initial model $f(\cdot; w_0)$ with parameters $w_0$. After $w_0$ was learned, a series of unlearning requests are received by streaming unlearning, and the $t$-th request comes with a forgetting data $F_t \subset D_0$. Upon receiving each request, the model needs to unlearn the corresponding $F_t$, resulting in an unlearned model $f(\cdot, w_t)$. When the forgetting data is accumulated through the series of forgetting data received, the size of the remaining data will decrease gradually. Upon receiving the $t$th request, the remaining data will defined recursively as $D_t = D_{t-1} - F_t$, and the $t$-th request comes with a forgetting data $F_t \subset D_{t-1}$.

We consider a stream data removal request $\{F_t\}_{t=1}^{T}$. To achieve the goal in Eq. 1 in the $t$-th round, we need to minimize the following objective, formulated as

$$\min_{w_t} \left( \mathcal{L}(D_t, w_t) - \mathcal{L}(D_t, w_t^*) + \lambda \mathcal{R}(F_t, w_t) \right), \tag{3}$$

where $w_t^*$ denotes the optimal parameters for the $t$-th round of forgetting and $\mathcal{L}(D_t, w_t^*)$ is a fixed value during unlearning. After all $T$ rounds, we define the regularized unlearning dynamic regret of the continuously updated model against the optimal models in terms of cumulative risks as the objective for the *online unlearning* paradigm

$$U\text{-}Regret_T = \sum_{t=1}^{T} \left( \mathcal{L}(D_t, w_t) - \mathcal{L}(D_t, w_t^*) + \lambda \mathcal{R}(F_t, w_t) \right). \tag{4}$$

In online unlearning, the objective differs from that of online learning. Rather than acquiring new knowledge from incoming data, the goal of online unlearning is to remove the knowledge associated with this data, leading to a significant reduction in the model performance on these data. This is why the unlearning regret, as shown in Eq. 5, includes an additional second term, $\lambda \mathcal{R}(F_t, w_t)$. Furthermore, in online unlearning, $D_t$ is gradually shrunk rather than expanded, as it would be in online learning. Additionally, since online unlearning begins with a well-trained model, which is often a deep model in modern practice, it is potentially risky to assume that the model will adhere to the convexity or pseudo-convexity assumptions typically made in online learning.

## 3 ONLINE UNLEARNING METHODOLOGY

To address the challenges in the online unlearning problem, we introduce the proposed SAFE algorithm in this section. In Subsection 3.1, we first design a novel online unlearning risk estimator to estimate the real risk in Eq. 4. The risk estimator includes the recorded training data risk, the recovered forgetting data risk, and the distribution shift risk. Next, the calculation process of the distribution shift is presented in Subsection 3.2. Then, we provide the SAFE algorithm in Subsection 3.4 with theoretical analysis in Subsection 3.5

### 3.1 ONLINE UNLEARNING RISK ESTIMATOR

To achieve the online unlearning objective in all rounds in Eq. 4, we first need to estimate the online unlearning risk in each round as shown in Eq. 5. Considering that the true risk in the $(t\text{-}1)$-th round of request of the model $f$ parameterized by $w$ is:

$$w = \arg\min_{w_t} \left( \mathcal{L}(D_t, w_t) - \mathcal{L}(D_t, w_t^*) + \lambda \mathcal{R}(F_t, w_t) \right), \tag{5}$$

since $\mathcal{L}(D_t, w_t^*)$ is the optimized loss on the remaining data $D_t$ in the retrained model and can be discarded in the following optimizations since it is a fixed value. Then the risk in the $t$-th round can be written as:

$$R_t(w) = \mathcal{L}(D_t, w) + \lambda \mathcal{R}(F_t, w),$$

$R_t(w)$ in the above equation is equivalent to the following risk when defining the regularizer to be $1/|F_t| \sum_{(\mathbf{x},y) \in F_t} d_{\text{KL}}(f(\mathbf{x};w), f(\mathbf{x};w_t^*))$:

$$R_t(w) = \underbrace{\frac{|D_0|}{|D_t|} R_0(w)}_{(a)} - \underbrace{\frac{1}{|D_t|} \sum_{i=1}^{t} \sum_{(\mathbf{x},y) \in F_i} \ell(f(\mathbf{x};w), y)}_{(b)} + \underbrace{\frac{\lambda}{|D_t|} \sum_{i=1}^{t} \sum_{(\mathbf{x},y) \in F_i} d_{\text{KL}}(f(\mathbf{x};w), f(\mathbf{x};w_t^*))}_{(c)},$$

$$\tag{6}$$

where Eq. 6 (a) presents the training risk on all training data $D_0$, Eq. 6 (b) denotes the training risk on cumulative forgetting data through all $t$ forgetting rounds, and Eq. 6 (c) stands for the regularizer term, showing the discrepancy between forgetting data predictions on the unlearned and retrained models.

In the unlearning procedure, we first need to estimate the forgetting data prediction $f(\mathbf{x}; w_t^*)$, where $w_t^* = \arg\min_w \mathcal{L}(D_t, w)$. However, $w_t^*$ cannot be obtained during unlearning, considering the remaining data $D_t$ cannot be frequently accessed in the reach round. Instead of estimating $w_t^*$, we estimate $f(\mathbf{x}; w_t^*)$ directly in order to further optimize Eq.6(c). The estimation of $f(\mathbf{x}; w_t^*)$, denoted as $\tilde{f}_i(\mathbf{x}; w_0)$, will be obtained through a distribution shift approach by analysing the distribution shift between $D_{t-1}$ and $D_t$. In the following subsection, we will introduce especially this distribution shift approach.

### 3.2 DISTRIBUTION SHIFT RISK

In the unlearning process, the optimal models in each unlearning round are those retrained on the remaining data $\{D_t\}_{t=1}^{T}$, which has removed the information of forgetting data $\{F_t\}_{t=1}^{T}$ in each unlearning round. Since the data in $D_t$ evolves with the ongoing unlearning of $F_t$, the corresponding distribution of $D_t$ will also shift in each round.

After $t$ rounds of the unlearning requests, the shifted prediction probability of $x$ on each class $y$ (i.e. $Q_t(y|\mathbf{x})$) from the initial prediction $Q_0(y|\mathbf{x})$ can be denoted as:

$$Q_t(y|\mathbf{x}) = \frac{Q_0(\mathbf{x})}{Q_t(\mathbf{x})} \frac{Q_t(y)}{Q_0(y)} \frac{Q_t(\mathbf{x}|y)}{Q_0(\mathbf{x}|y)} Q_0(y|\mathbf{x}) \propto \frac{Q_t(y)}{Q_0(y)} \frac{Q_t(\mathbf{x}|y)}{Q_0(\mathbf{x}|y)} Q_0(y|\mathbf{x}). \tag{7}$$

In the above equations, We assume that the feature marginal distributions $Q_t(\mathbf{x})$ remain unchanged in the same $t$-th round, and therefore, $Q_t(\mathbf{x})/Q_0(\mathbf{x})$ will be constant. $Q_t(y)/Q_0(y)$ is the ratio of the proportions of the data belonging to class $y$ in $D_0$ and $D_t$, which are denoted as $D_0^{[y]}$ and $D_t^{[y]}$

where the superscript $[y]$ stands for the data belonging to the class $y$. Then, $Q_t(\mathbf{x}|y)$ and $Q_0(\mathbf{x}|y)$ represent the data distribution conditioned on each class $y$.

To effectively estimate $Q_t(\mathbf{x}|y)$ and $Q_0(\mathbf{x}|y)$, we can approximate the conditional distribution as a Gaussian, i.e., $z_t(\mathbf{x}|y) = N(\mu_t^{[y]}, \mathbf{\Sigma}_t^{[y]})$ where $\mu_t^{[y]}$ and $\mathbf{\Sigma}_t^{[y]}$ stand for the mean vector and covariance of the low-dimensional vectors of the $D_t$ of class $y$ in the $t$-th round. Note that approximating data as a Gaussian Distribution is a common approximation used in many learning methods, such as in Variational Autoencoders (VAEs) (Kingma & Welling, 2014) and Bayesian Neural Networks (Neal, 2012). At the beginning of the unlearning process, the original data $\mathbf{x}$ is projected into another space through a series of linear transformations, which standardize the data to ensure that the projected vector $v^{[y]}(\mathbf{x})$ follows a normalized Gaussian distribution conditioned on $y$. In each round, the distribution of $D_t$ will be influenced by the removal of the forgetting data. Given that the size of the forgetting data is relatively small compared to the remaining data, the Gaussian distribution can still be approximately maintained, though the mean and covariance may undergo slight adjustments. Specifically, the mean vector can be derived by:

$$\mu_t^{[y]} = \frac{|D_{t-1}^{[y]}|}{|D_t^{[y]}|}\mu_{t-1}^{[y]} - \tilde{\mu}_t^{[y]}, \tag{8}$$

and

$$\mathbf{\Sigma}_t^{[y]} = \frac{|D_t^{[y]}| - 1}{|D_{t-1}^{[y]}| - 1}\mathbf{\Sigma}_{t-1}^{[y]} - \tilde{\mathbf{\Sigma}}_t^{[y]} + c(\mu_t^{[y]}, \mu_{t-1}^{[y]}). \tag{9}$$

In the above equations, $\tilde{\mu}_t^{[y]} = 1/|F_t^{[y]}| \sum_{x \in F_t^{[y]}} v^{[y]}(\mathbf{x})$ and $\tilde{\mathbf{\Sigma}}_t^{[y]} = \frac{1}{|D_t^{[y]}|-1}[\sum_{x \in F_t^{[y]}}(v^{[y]}(\mathbf{x}) - \tilde{\mu}_t^{[y]})(v^{[y]}(\mathbf{x}) - \tilde{\mu}_t^{[y]})^T]$ are the mean and covariance of the Gaussian vectors of the forgetting data $F_t^{[y]}$ that belongs to class $y$. Then $c(\mu_t^{[y]}, \mu_{t-1}^{[y]}) = |D_{t-1}^{[y]}|(\mu_{t-1}^{[y]} - \mu_t^{[y]})(\mu_{t-1}^{[y]} - \mu_t^{[y]})^T - |F_t^{[y]}|(\tilde{\mu}_t^{[y]} - \mu_t^{[y]})(\tilde{\mu}_t^{[y]} - \mu_t^{[y]})^T]$.

In the unlearning process, the updating in Eqs. 8 and 9 can be done incrementally. We only need to calculate the corresponding $z_t(\mathbf{x}|y)$ for the forgetting data through the recorded projection to update $\mu_t^{[y]}$ and $\mathbf{\Sigma}_t^{[y]}$. Apart from the initial statistics of training data to get $\mu_0^{[y]}$ and $\mathbf{\Sigma}_0^{[y]}$, we *do not require any access to the original training data*. After obtaining the Gaussian Class-conditional probability $z_t(\mathbf{x}|y) = N(\mathbf{x}; \mu_{t+1}, \mathbf{\Sigma}_{t+1})$, we can incorporate $z_t(\mathbf{x}|y)$ into Eq. 7 to get the shifted feature-conditioned distribution:

$$Q_t(y|\mathbf{x}) = q_t^{[y]}(\mathbf{x})Q_0(y|\mathbf{x}) \propto \frac{Q_t(y)}{Q_0(y)}\frac{z_t(\mathbf{x}|y)}{z_0(\mathbf{x}|y)}Q_0(y|\mathbf{x}), \tag{10}$$

where we use softmax to normalize the values of $Q^t(y|\mathbf{x})$ for all $y$ to control that $\sum_y Q_t(y|\mathbf{x}) = 1$ and $q_t(\mathbf{x})$ is an vector with the same dimension of output $f(\mathbf{x}; w_0)$ and $q_t^{[y]}(\mathbf{x}) \propto \frac{Q_t(y)}{Q_0(y)}\frac{z_t(\mathbf{x}|y)}{z_0(\mathbf{x}|y)}$.

$Q_t(\mathbf{x}|y)$ can also be approximated by other distributions, such as the $\chi^2$ and $t$ distributions. However, when using these alternative distributions, estimating the distribution parameters incrementally, as we do for $\mu_t$ and $\Sigma_t$ in Eqs.8 and 9, becomes significantly more challenging. For this reason, we adopt the Gaussian distribution in our method. We also empirically test the deviation of $Q(\mathbf{x}|y)$ from the Gaussian distribution, and results in Appendix C suggest that the transformed vectors satisfy the Gaussian distributions under Mardia's test (Mardia, 1970).

### 3.3 Online Unlearning Optimization

After obtaining the shifted distribution for each round, we derive the reference probability for the predictions of the forgetting data: $\tilde{f}_t(\mathbf{x}; w_0^*) = q_t(\mathbf{x})f(\mathbf{x}; w_0)$ and the forgetting data prediction $f(\mathbf{x}; w_t^*)$ can be estimated by $\tilde{f}_t(\mathbf{x}; w_0^*)$. The estimated risk for the online unlearning problem is:

$$\hat{R}_t(w) = \frac{|D_0|}{|D_t|}R_0(w) - \frac{1}{|D_t|}\sum_{i=1}^{t}\sum_{(\mathbf{x},y) \in F_i}(\ell(f(\mathbf{x}; w), y) - \lambda d_{\mathrm{KL}}(f(\mathbf{x}; w), q_t(\mathbf{x})f(\mathbf{x}; w_0))). \tag{11}$$

The risk estimator of Eq. 11 can be proved to be equivalent to the population risk of Eq. 6 in each round of the unlearning. The detailed proof is provided in Appendix A.3, and the correctness of the condition in Theorem 1 has been proved in Lemma 1 in (Yu et al., 2018).

**Theorem 1.** *If* $\sqrt{|D_t|} \gg \sum_{i=1}^{t} |F_i|$ *and* $f(\mathbf{x}; w_0) = Q_0(y|\mathbf{x})$, *then* $\hat{R}_t(w)$ *is equivalent to* $R_t(w)$.

To calculate the estimated risk incrementally, we first record the initial gradient $\nabla R_0(w_0)$ on $D_0$. Then, during unlearning, we do not require $D_0$ anymore for $\hat{R}_t(w)$ and we calculate the distribution shift risk $\frac{1}{|D_t|} \sum_{i=1}^{t} \sum_{(\mathbf{x},y) \in F_i} (\ell(f(\mathbf{x}; w_0), q_t(\mathbf{x})f(\mathbf{x}; w_0)))$ and population risk $\frac{1}{|D_t|} \sum_{i=1}^{t} \sum_{(\mathbf{x},y) \in F_i} \ell(f(\mathbf{x}; w_0), y)$ through the accumulated forgetting data. Then, we can obtain $\hat{R}^t(w_0)$ and get the corresponding gradients. Next, we update the model by first-order optimization

$$w_t = w_0 - \gamma \frac{\nabla \hat{R}_t(w_0)}{||\nabla \hat{R}_t(w_0)||} = w_{t-1} - \gamma \nabla_t^{\text{step}}, \tag{12}$$

where $\gamma$ is the learning rate and $\nabla_t^{\text{step}} = (\frac{\nabla \hat{R}_t(w_0)}{||\nabla \hat{R}_t(w_0)||} - \frac{\nabla \hat{R}_{t-1}(w_0)}{||\nabla \hat{R}_{t-1}(w_0)||})$ and $|| \cdot ||$ stands for the $L_2$ norm.

---

**Algorithm 1** SAFE Algorithm

---

**Input** $D_0$, $\{F_t\}_{t=1}^{T}$, $w_0$, $\nabla R_0(\cdot)$, $\gamma$;
**Output** $\{w_t\}_{t=1}^{T}$;
1: **procedure** SAFE:
2:     Initial $w_0$ as model parameters before unlearning;
3:     Calculate low-dimension projectors and initial Gaussian parameters $\mu_0$ and $\Sigma_0$;
4:     Calculate the initial risk $R_0(w_0)$ on $D_0$;         // $D_0$ *will be dropped out.*
5:     **for** $t = 1, \ldots, T$ **do**:
6:         Estimate $\mu_t$ and $\Sigma_t$;
7:         Calculate the shift distribution risk and population risk for forgetting data;
8:         Calculate $\nabla \hat{R}_t(w^0)$;
9:         $w_t \leftarrow w_{t-1} - \gamma \nabla_t^{\text{step}}$;
10:     **end for**
11: **end procedure**

---

### 3.4 SAFE ALGORITHM

We provide the pseudo-code of the overall SAFE in Algorithm 1. Before unlearning starts, SAFE calculates the initial mean vector $\mu_0^{[y]}$ and covariance matrix $\Sigma_0^{[y]}$, and the initial risk $R_0(w_0)$ on the original training data. Then, in the unlearning procedure, we only need the specified forgetting data $\{F_t\}_{t=1}^{T}$ without any requirements on the remaining data $\{D_t\}_{t=0}^{T}$.[1]

We begin by calculating the distribution shift risk and the true risk of forgetting data on the initial model weights $w_0$. From this, we derive the estimated risk $\hat{R}_t(w_0)$ and the corresponding gradients $\nabla \hat{R}_t(w_0)$. By recording the model weights $w_{t-1}$ and gradients $\nabla \hat{R}_{t-1}(w_0)$ in the previous unlearning round, the model weights are updated by Eq. 12. Consequently, SAFE outputs a series of updated models with weights $\{w_t\}_{t=1}^{T}$, fulfilling the unlearning requests $\{F_t\}_{t=1}^{T}$.

In each unlearning round, the primary algorithmic time consumption occurs during distribution shift inference and the gradient update for $\hat{R}_t(w_0)$. Since the computational load for these stages remains constant regardless of the number of unlearning rounds, the time consumption per round is constant, resulting in a linear increase in total time consumption with the number of rounds. Regarding memory consumption, the storage requirements for distribution shift inference variables and model gradients remain unchanged in each unlearning round, leading to a constant total memory consumption across all unlearning requests.

---

[1]The implemented code will be made publicly available after the notification is released.

### 3.5 THEORETICAL GUARANTEE

In this section, we provide theoretical guarantees for the SAFE algorithm on the error bound between the performances of the unlearned and retrained models for the single round and cumulative rounds. The error bounds guarantee the performance closeness of the unlearned and retrained model on both remaining data and forgetting data.

**Theorem 2.** *If the risk $R_t(w)$ satisfies the upper-bounded gradient assumption with upper bound $U$, and the model weights satisfy the gradient assumption with upper bound $W$ (i.e. $|w| \leq W$ $||\nabla R_t(w)|| \leq U$). For any sequence of unlearning requests $\{F^t\}_1^T$ with the rounds to be $T$, we set $\gamma = \frac{\sqrt{W}}{4\sqrt{T}}$. Then, by applying the first-order optimization algorithm:*
*(i) the error in the $t$-th rounds of unlearning compared with the optimal model state $w_t^*$ is bounded:*

$$\mathbb{E}\left[R_t(w_t) - R_t(w_t^*)\right] \leq O(\sqrt{T}).$$

*(ii) the accumulated unlearning regret across all requests is bounded:*

$$\mathbb{E}\left[U\text{-}Regret_T(\{w_t\}_1^T)\right] \leq O(V_T\sqrt{T} + \Delta_T),$$

*where $V_T = 1 + \sum_{t=1}^T ||w_t^* - w_{t-1}^*||$, $\Delta_T = 2\sum_{t=1}^T div(D_t, F_t)$, and $div(D_t, F_t)$ denotes the divergence between the distribution of remaining data $D_t$ and forgetting data $F_t$ in the $t$-th round of unlearning as defined in (Ben-David et al., 2006).*

Based on the above theorem, the upper bound error between the retrained model and unlearned model and the total unlearning regret consists of both the dynamic regret part $O(V_T\sqrt{T})$ and distribution shift part $\Delta_T$. When the forgetting data are uniformly sampled from the training dataset, $\Delta_T$ makes a minor influence on the unlearning performance, and SAFE can reach a lower unlearning regret. However, when the forgetting data and the remaining data have two separate distributions, like the data in different classes, $\Delta_T$ will lead to a higher upper bound error and then lead to the unlearning regret.

For the dynamic regret part $O(V_T\sqrt{T})$, to the best of our knowledge, the proved error bound is the first error bound of dynamic regret for the streaming unlearning problem. However, we acknowledge that there is room for improvement in the current bound of $O(V_T\sqrt{T})$. Previous works in online learning have achieved better dynamic regret bounds of $O(\sqrt{V_T T})$ (Gao et al., 2018) and $O(T^{\frac{2}{3}})$ (Ghai et al., 2022) under similar conditions without the restriction of convexity. This difference may inspire future works to establish a better bound for online unlearning. In addition, the previous batch machine unlearning algorithm Sekhari et al. (2021) achieves an upper bound of $O(m^2)$, where $m$ stands for the size of the forgetting data in the single batch unlearning. $m$ usually increases linearly to the size of the unlearning request $T$ in the whole stream unlearning settings. Therefore, it can achieve an upper bound of $O(T^2)$ in the stream manner. In comparison, the error bound $O(V_T\sqrt{T} + \Delta_T)$ of SAFE is much smaller.

## 4 EXPERIMENTS

### 4.1 EXPERIMENT SETTINGS

**Datasets and Models**: To validate the effectiveness of LAF, we conduct experiments on four datasets: **CIFAR10 Features**, **DIGITS** (MNIST) (LeCun, 1998), **FASHION** (Fashion MNIST) (Xiao et al., 2017), and raw **CIFAR10** (Krizhevsky et al., 2009). On CIFAR10 Features dataset, we choose a **ResNet-18** to extract features and use a logistic regression model (**LR**) for binary classifications. On the two MNIST datasets, we use a two-layer convolutional neural network (**CNN**) (LeCun et al., 1995), while on the CIFAR10 dataset, we choose a **ResNet-18** backbone (He et al., 2016). In Appendix B.1, we show the details about data pre-processing and model structures. For the hyperparameters, we set $\lambda = 2000$ and tune $\gamma$ for better unlearning results, which is shown in Appendix B.6.

**Baselines**: We compare the performance of the SAFE algorithm with Retrain which represents the standard results from the retraining models, two other unlearning work to handle streaming unlearning requests **without remaining data**: LCODEC (Mehta et al., 2022) and Descent (Neel et al., 2021), and two other methods **requiring remaining data**: Unrolling (Thudi et al., 2022), and CaMU (Shen et al., 2024a). All the experiments on these baselines are conducted under 10 random seeds.

**Evaluations**: For effectiveness evaluations, we assess the unlearning algorithm using five metrics: **RA**(Remaining Accuracy), **UA**(Unlearning Accuracy), and **TA**(Test Accuracy), which denote the prediction accuracy of the post-unlearning model on the remaining data, forgetting data, and test data. The closer value to the retrained model indicates better unlearning performance for these metrics. We also compare **JS**(Jensen–Shannon Divergence), which stands for the Jensen–Shannon divergence. It measures the divergence between the outputs of the model after unlearning and the model retrained on the remaining data. A lower JS indicates smaller differences between the two models and, therefore, better performance of the unlearning method. **Remain JS**, **Forget JS**, and **Text JS** stand for the JS comparisons on remaining, forgetting, and test data, respectively. We also check the **ASR**, the attack accuracy of the MIA (Shokri et al., 2017; Chen et al., 2021) and we use the same MIA evaluations as (Fan et al., 2024; Jia et al., 2023).

Table 1: Complete comparison results in 20 rounds of unlearning, which remove 400 data points (avg%±std%). The **bold** record indicates the best, and the underlined record indicates the second-best.

| Method | RA | UA | TA | Remain JS | Forget JS | Test JS | ASR |
|---|---|---|---|---|---|---|---|
| | | | | **MNIST** | | | |
| Retrain | 99.68±0.05 | 98.89±0.09 | 99.00±0.05 | 0.00±0.00 | 0.00±0.00 | 0.00±0.00 | 79.25±1.14 |
| Unroll (Thudi et al., 2022) | 99.24±0.22 | **98.91±0.15** | 98.61±0.19 | 0.59±0.11 | 0.88±0.11 | 0.89±0.10 | 79.27±1.15 |
| CaMU (Shen et al., 2024a) | 98.94±0.36 | 98.72±0.79 | 98.54±0.42 | 12.12±2.29 | 13.43±1.56 | 11.98±2.22 | 79.05±1.13 |
| LCODEC (Mehta et al., 2022) | 96.26±1.95 | 96.27±1.88 | 95.60±1.94 | 2.61±1.13 | 2.87±1.25 | 2.93±1.25 | 78.79±2.22 |
| Descent (Neel et al., 2021) | 98.78±0.53 | 98.72±0.53 | 98.27±0.46 | 0.89±0.32 | 1.05±0.31 | 1.13±0.28 | 79.24±1.14 |
| SAFE | **99.74±0.03** | 99.63±0.05 | **99.04±0.02** | **0.34±0.03** | **0.61±0.06** | **0.65±0.03** | **79.26±1.15** |
| | | | | **MNIST Fashion** | | | |
| Retrain | 96.44±0.15 | 90.76±0.33 | 90.40±0.15 | 0.00±0.00 | 0.00±0.00 | 0.00±0.00 | 79.57±0.51 |
| Unroll (Thudi et al., 2022) | 90.61±0.91 | 89.08±0.94 | 87.99±0.85 | 4.21±0.52 | 4.75±0.56 | 4.82±0.53 | **79.14±0.65** |
| CaMU (Shen et al., 2024a) | 91.32±0.36 | 90.45±0.83 | 89.00±0.40 | 10.40±0.45 | 11.53±0.43 | 10.51±0.41 | 78.07±0.53 |
| LCODEC (Mehta et al., 2022) | 86.21±4.27 | 86.26±4.15 | 82.28±3.89 | 7.81±2.57 | 8.60±2.38 | 8.94±2.45 | 77.41±1.75 |
| Descent (Neel et al., 2021) | 89.52±1.38 | 89.27±1.54 | 87.40±1.10 | 4.81±0.75 | 4.99±0.73 | 5.19±0.67 | 78.28±0.77 |
| SAFE | **91.73±0.25** | **90.60±0.43** | **89.04±0.30** | **3.62±0.15** | **4.09±0.25** | **4.20±0.17** | 77.96±0.49 |
| | | | | **CIFAR10 Feature** | | | |
| Retrain | 85.61±0.13 | 84.16±0.57 | 85.01±0.16 | 0.00±0.00 | 0.00±0.00 | 0.00±0.00 | 48.72±1.95 |
| Unrolling (Thudi et al., 2022) | 86.25±0.06 | 84.97±0.27 | 85.51±0.03 | 0.06±0.02 | 0.07±0.02 | 0.06±0.02 | 49.32±1.92 |
| CaMU (Shen et al., 2024a) | 85.77±0.12 | **83.40±0.62** | **84.99±0.15** | 0.12±0.03 | 0.12±0.03 | 0.13±0.03 | 51.50±1.62 |
| LCODEC (Mehta et al., 2022) | 85.98±0.06 | 84.91±0.27 | 85.34±0.03 | **0.05±0.02** | **0.06±0.02** | **0.06±0.02** | **48.82±1.88** |
| Descent (Neel et al., 2021) | 86.24±0.07 | 84.94±0.30 | 85.50±0.01 | 0.07±0.02 | 0.07±0.02 | 0.07±0.02 | 49.45±1.91 |
| SAFE | **85.74±0.13** | 84.10±1.15 | 84.94±0.14 | 0.13±0.03 | 0.14±0.04 | 0.13±0.04 | 49.38±2.38 |
| | | | | **CIFAR10** | | | |
| Retrain | 97.61±0.25 | 91.78±0.49 | 91.19±0.34 | 0.00±0.00 | 0.00±0.00 | 0.00±0.00 | 64.38±1.34 |
| Unroll (Thudi et al., 2022) | 93.59±2.57 | 90.46±1.92 | 86.97±2.12 | 4.02±1.38 | 5.96±1.09 | 5.82±1.27 | 74.96±2.95 |
| CaMU (Shen et al., 2024a) | **95.71±1.09** | 93.13±3.32 | **89.58±1.09** | 4.96±0.82 | 7.38±2.95 | 6.14±0.85 | 74.54±2.95 |
| LCODEC (Mehta et al., 2022) | 23.95±3.71 | 24.08±4.18 | 23.33±3.54 | 49.21±2.62 | 48.93±8.42 | 48.81±2.85 | 54.47±8.42 |
| Descent (Neel et al., 2021) | 74.01±27.50 | 74.24±27.58 | 70.01±25.34 | 17.63±17.06 | 17.66±16.6 | 17.66±16.4 | **71.95±7.67** |
| SAFE | 94.22±0.85 | **92.30±0.45** | 87.74±0.84 | **3.46±0.46** | **5.67±0.28** | **5.37±0.28** | 74.18±3.31 |

## 4.2 PERFORMANCE COMPARISON

First, we evaluate the unlearning performance of long sequential requests of the proposed SAFE algorithm. Table 1 shows the average performance comparisons over 20 rounds of requests, each requiring the removal of 400 randomly selected data points. The results demonstrate that the SAFE algorithm can achieve the closest average results to the retrained model on the two MNIST and CIFAR10 datasets and the second closest average results on the CIFAR10 Feature, as shown in the GAP column. Specifically, in terms of accuracy, SAFE achieves nearly all the best results on the two MNIST datasets. Although SAFE shows higher forgetting data accuracy on MNIST compared to others, it achieves the smallest JS Divergence on the forgetting data, indicating that SAFE can produce output performances for each instance that are closest to those of the retrained model. Among all the methods, Descent and SAFE realize the requests in the stream unlearning manner, while the other four baselines conduct the batch unlearning on the accumulated forgetting data. For the CIFAR10 Feature dataset, where a simple logistic regression model with convex loss is used, the advantage of online unlearning is not that significant due to the lack of reliance on any convexity assumption.

However, SAFE still achieves nearly all the top results. On the CIFAR10 dataset, where a more complex ResNet model is used, preserving performance after unlearning is challenging. Even so, SAFE achieves the second-highest results in maintaining remaining data accuracy and test accuracy while also achieving the best unlearning performance.

We also conducted statistical testing on all the experimental results presented in Table 1. We compare our proposed method with the other methods based on a two-tailed t-test with a 90% confidence level. The results of RA, UA, TA, Remaining JS, and Test JS on the CIFAR-10 are significantly better than other baselines. Additionally, the results of RA, TA, and all JS metrics on the MNIST and FMNIST datasets are also significantly better than the other methods. These results demonstrate that the proposed method performs significantly better than the other methods.

Apart from the performance analysis on accuracy and JS divergence evaluation, we provide the analysis of MIA in the following. Then, our proposed method achieves the best performance on the MNIST dataset and the second-best results on CIFAR-10, demonstrating its effectiveness on complex datasets. For the other two datasets, Fashion and CIFAR-10 Feature, our method achieves middle-tier performance but remains comparable to other baselines. Compared with CAMU and Unroll, the gradient ascent methods LCODEC, Descent, and SAFE achieve a relatively lower MIA compared with the retrained model due to the fewer update steps on the model during unlearning. It will have less impact on the model parameters and prediction results.

Then Figure 1 demonstrates the performance changes with the increasing of unlearning rounds. The results of SAFE are in blue lines, and the results of the retrained model are in red lines. Although in Figure 1(a), the remaining data accuracy of SAFE is lower than that of the retrained model, it is still the closest among all methods. Additionally, in Figure 1(c), SAFE maintains high prediction performance in all 20 rounds of test accuracy. This performance is comparable to CaMU, which requires remaining data to preserve performance, whereas SAFE does not require unrelated training data. For the forgetting data performance, Figure 1(b) demonstrates the effectiveness of removing forgetting data information. Both the group-level accuracy and instance-level divergence show that SAFE can achieve results closest to those of the retrained model on forgetting data. To further evaluate the SAFE algorithm, we present experimental results under different request settings in Appendix B.5, an analysis of the effect of learning rate in Appendix B.6, and ablation studies in Appendix B.7.

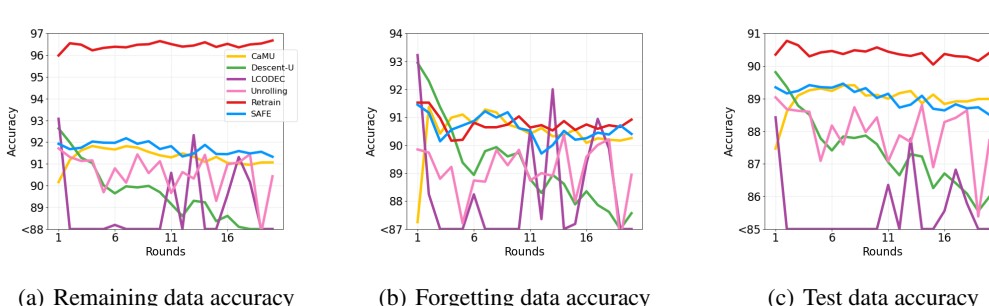

(a) Remaining data accuracy      (b) Forgetting data accuracy      (c) Test data accuracy

Figure 1: Model performance against unlearning rounds on MNIST Fashion. The red line stands for the performance changes of the retrained model and the blue line stands for the model after unlearning via SAFE

### 4.3 EFFICIENCY ANALYSIS

In the efficiency comparisons, we recorded the average time cost for 20 rounds of unlearning, with the results presented in Figure 2. SAFE achieved average time costs of 0.59 and 0.55 seconds on the two MNIST datasets, while the second-fastest algorithm, L-CODEC, required 1.66 and 1.14 seconds, more than twice the time of SAFE. On the CIFAR10 dataset, the advantage of SAFE is even more pronounced. SAFE required only 1.57 seconds per request, whereas the second-fastest algorithm, Descent, required 12.90 seconds, which is nearly nine times slower than SAFE. On the CIFAR10 Feature dataset, L-CODEC achieved the highest efficiency at 0.31 seconds due to the simpler model structure, with SAFE achieving the second-highest efficiency at 0.44 seconds. These time efficiency results highlight the leading advantage of SAFE in handling sequential requests,

especially for complex models such as CNNs and ResNet. As for the memory cost, we provide an analysis in Appendix B.8,

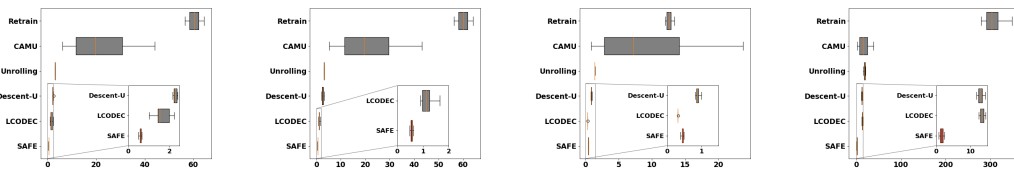

(a) Time cost in MNIST  (b) Time cost in MNIST Fashion  (c) Time cost in CIFAR10 Feature  (d) Time cost in CIFAR10

Figure 2: Time cost comparisons for stream instance unlearning. The x-axis stands for the time (seconds) used to realize each round of unlearning in the stream settings

Table 2: Ablation results for stream instance unlearning(avg%±std%).

| Method | RA | UA | TA | JS | GAP | RA | UA | TA | JS | GAP |
|---|---|---|---|---|---|---|---|---|---|---|
| | | | **MNIST** | | | | | **MNIST-Fashion** | | |
| Retrain | 99.68±0.05 | 98.89±0.09 | 99.00±0.05 | 0.00±0.00 | 0.00 | 96.44±0.15 | 90.76±0.33 | 90.40±0.15 | 0.00±0.00 | 0.00 |
| w/o DSR | 99.81±0.01 | 99.78±0.02 | 99.06±0.01 | **0.57±0.05** | 0.41 | **92.23±0.01** | 92.00±0.32 | **89.58±0.01** | **3.65±0.16** | 2.48 |
| w/o FR | 99.75±0.02 | 99.65±0.05 | 99.05±0.02 | 0.61±0.05 | 0.37 | 91.85±0.21 | **90.80±0.38** | 89.16±0.24 | 4.00±0.22 | 2.47 |
| w/o TR | 99.54±0.01 | **99.39±0.07** | 98.88±0.01 | 0.71±0.04 | 0.37 | 89.38±0.32 | 87.47±0.60 | 86.86±0.30 | 5.48±0.19 | 4.84 |
| SAFE | **99.74±0.03** | 99.63±0.05 | **99.04±0.02** | 0.61±0.12 | 0.36 | 91.73±0.25 | 90.60±0.43 | 89.04±0.30 | 4.09±0.25 | 2.58 |
| | | | **CIFAR10 Feature** | | | | | **CIFAR10** | | |
| Retrain | 85.61±0.13 | 84.16±0.57 | 85.01±0.16 | 0.00±0.00 | 0.00 | 97.61±0.25 | 91.78±0.49 | 91.19±0.34 | 0.00±0.00 | 0.00 |
| w/o DSR | 85.89±0.03 | 85.87±0.41 | 85.10±0.03 | 0.15±0.05 | 0.38 | **97.35±0.01** | 97.44±0.10 | **90.84±0.01** | **3.96±0.19** | 2.56 |
| w/o FR | 85.88±0.03 | 85.38±0.82 | **85.07±0.08** | **0.13±0.03** | 0.43 | 94.42±0.76 | 92.59±0.39 | 87.92±0.76 | 5.55±0.24 | 2.71 |
| w/o TR | 85.31±1.08 | 82.34±3.24 | 84.68±7.40 | 0.18±0.03 | 0.82 | 92.30±1.05 | 89.36±0.66 | 86.03±1.04 | 6.85±0.38 | 4.94 |
| SAFE | **85.74±0.13** | **84.10±1.15** | 84.94±0.14 | 0.14±0.04 | 0.30 | 94.22±0.85 | **92.30±0.45** | 87.74±0.84 | 5.67±0.28 | 3.26 |

## 4.4 ABLATION STUDY

Table 2 presents the results of the ablation study for the proposed algorithm, where we sequentially remove the distribution shift loss (**DSR**), the forgetting data gradient (**FR**), and the initial training data gradient (**TR**). First, when the distribution shift loss is removed, the forgetting data accuracies on all four datasets are similar to the original forgetting data accuracies, implying that the information of the streaming forgetting data has not been effectively removed from the model. Second, when only the forgetting data gradient is removed, the forgetting data accuracies approach those of the retrained models. However, the results of the complete SAFE still outperform those without the forgetting data gradient. Lastly, when the initial training data gradient is removed, there is a significant drop in accuracies across all datasets.

These experimental results indicate that the distribution shift loss and the forgetting data gradient contribute significantly to the unlearning process. The distribution shift loss is the dominant factor, while the forgetting data gradient also provides a substantial contribution. Additionally, the initial training data gradient is crucial in maintaining overall performance.

## 5 CONCLUSION

In this paper, we address the practical requirements of long sequential unlearning by introducing an online unlearning paradigm. This paradigm is designed to realize sequential unlearning requests with high forgetting accuracy and efficiency. We first conceptualize unlearning as the distribution shift problem and estimate the Multivariate Gaussian distribution of low-dimensional vectors of the training data of each class. Then, we propose a novel SAFE (**S**tream-**A**ware **F**org**e**tting) algorithm alongside a first-order optimization that can reach a low regret bound. We conducted extensive experiments and the results show that SAFE consistently achieves top or near-top performances across various evaluations, including more than double the time efficiency compared with the second-most efficient algorithm, demonstrating its clear advantages over other baseline methodologies.

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

## A    APPENDIX

### A.1    RELATED WORK

#### A.1.1    MACHINE UNLEARNING

Machine unlearning requires the removal of information of forgetting data from the original model while preserving the knowledge contained in the remaining data (Bourtoule et al., 2021; Xu et al., 2024). Current research on machine unlearning can be categorized into two primary branches based on unlearning requests: *batch unlearning* and *streaming unlearning*. Batch data unlearning focuses on removing a specific data group within the same batch (Bourtoule et al., 2021; Thudi et al., 2022; Chundawat et al., 2023a; Chen et al., 2023; Kurmanji et al., 2023; Shen et al., 2024a;b). This approach typically requires access to the original training data and fine-tuning it to maintain high prediction performance. For instance, (Bourtoule et al., 2021) proposes retraining the model using small data shards from the remaining dataset and ensembling the final results for increased efficiency. Similarly, (Thudi et al., 2022) performs incremental training with the forgotten data in the first batch, recording gradients during the initial batch and adding these recorded gradients to the weights after incremental training. In contrast, streaming unlearning addresses continuous data removal requests (Gupta et al., 2021; Li et al., 2021; Neel et al., 2021; Chien et al., 2024). For example, (Gupta et al., 2021) extends (Bourtoule et al., 2021) to be more adaptive to incremental and decremental learning requests in a streaming context. (Neel et al., 2021) proposes a perturbed gradient-descent algorithm on data partitions to update models for stream unlearning requests. (Chien et al., 2024) fine-tunes the model with noisy gradients for unlearning, which can be extended to streaming unlearning with limited error increase. However, these approaches still face limitations. Some are restricted to convex loss functions (Neel et al., 2021), while others still rely on full training data access and retraining throughout the unlearning process (Bourtoule et al., 2021; Chien et al., 2024). These divergent methodologies underscore the challenges of efficiently applying machine unlearning across various data types and model structures (Gupta et al., 2021; Li et al., 2021; Neel et al., 2021; Chien et al., 2024).

#### A.1.2    ONLINE LEARNING

Online learning focuses on the learning task of a continuous data stream, which regards the minimization of regret risk as the objective (Hoi et al., 2021). With the increasing size of the incoming data, the training data distribution will face a significant shift. Therefore, adapting to the shifted distribution effectively and efficiently becomes one of the main tasks of online learning. Among these works, the extensions of the regret risk become a requisite research. The basic regret compares the cumulative risks among models in each unlearning step with the global optimal model after learning all data in the stream (Zinkevich, 2003b); the adaptive regret reduces the time length of the whole stream into smaller time windows and compares the cumulative risks with the optimal models in different time windows (Hazan et al., 2006); the dynamic regret directly compares the updated models with the optimal ones for each learning request (Zinkevich, 2003a; Besbes et al., 2015; Zhang et al., 2018). In addition, optimizing the regret risks determines the effectiveness and efficiency of the online learning algorithm. Specifically, the optimization methods include Online Gradient Descent (OGD) (Zinkevich, 2003b) for the first-order optimization and Online Newton Step (ONS) (Hazan et al., 2007) for the second-order optimization. In addition, Online Mirror Descent (OMD) (Duchi et al., 2010) is also a common approach, which generalizes OGD to perform updates in the dual space, which can be transformed through a regularize. Noticed that under different types of regret, different optimization methods can reach different rigorous error bounds. Last but not least, in practical problems like label shift problems, several online learning algorithms have been proposed (Wu et al., 2021; Bai et al., 2022; Baby et al., 2023), which connect the label shift with online learning via continuously updated classification margins. All the algorithm algorithms focus on the learning tasks with data stream while there is still a research gap in exploring online algorithms for the streaming unlearning request.

Online unlearning differs from online learning in the following aspects: Firstly, in machine unlearning, the initial model already has comprehensive knowledge about all the training data, including the data to be forgotten. Online unlearning aims to remove the information about the forgotten data, whereas online learning focuses on learning from newly incoming data. This makes the unlearning process inherently more challenging than the learning process. Secondly, the availability of training data in

online learning and online unlearning differs. In online unlearning, the size of the remaining data progressively decreases, and the models must adapt to new optimal states based on this remaining data. However, online unlearning does not provide access to the remaining data and must be addressed based on the provided forgetting data. Thirdly, from a practical standpoint, our online unlearning method, SAFE, does not assume any convexity or pseudo-convexity for the training loss. We only assume bounded weights and gradients on the training data. These assumptions are more practical as the initial model in unlearning problems has been well-trained, and the gradients on the current training data have stabilized.

## A.2 NOTATION

We provide a table of all notations of the main paper in Table 3.

Table 3: Table of Notation

| Notation | Explanation |
|---|---|
| $A$ | Learning algorithm |
| $U$ | Unlearning algorithm |
| $t$ | Round number of unlearning request |
| $T$ | Total length of unlearning request |
| $D_0$ | Original training data |
| $D_t$ | Remaining data in the $t$-th round of request |
| $F_t$ | Forgetting data in the $t$-th request |
| $f(\cdot;\cdot)$ | Prediction model |
| $\mathcal{L}(\cdot,\cdot)$ | Training loss |
| $\mathcal{R}(\cdot,\cdot)$ | Regularizer |
| $\ell(\cdot,\cdot)$ | Classification loss function |
| $d_{\mathrm{KL}}(\cdot,\cdot)$ | KL divergence loss |
| $w_t$ | Updated parameters in the $t$-th round of request |
| $w_t^*$ | Optimal parameters in the $t$-th round of request |
| $\mu_t^{[y]}, \mathbf{\Sigma}_t^{[y]}$ | Mean and covariance matrix estimated Gaussian distribution of remaining data for class $y$ in the $t$-th round of request |
| $\tilde{\mu}_t^{[y]}, \tilde{\mathbf{\Sigma}}_t^{[y]}$ | Mean and covariance matrix estimated Gaussian distribution of forgetting data for class $y$ in the $t$-th round of request |
| $z(\mathbf{x}\|y)$ | Estimated Gaussian distribution in the $t$-th round of request |
| $R_t(\cdot)$ | True risk in the $t$-th round of request |
| $\hat{R}_t(\cdot)$ | Risk estimator in the $t$-th round of request |
| $\nabla(\cdot)$ | Gradient of the function |
| $\|\cdot\|$ | L2 norm |
| $\gamma$ | Learning rate in each update |
| $W$ | Upper bound of the norm of model parameters |
| $U$ | Upper bound of the norm of gradients |

## A.3 THEORITICAL PROOF

**Lemma 1.** *(**Berry-Esseen theorem**) Let $\mathbf{X}_1, \mathbf{X}_2, \ldots, \mathbf{X}_n$ be independent and identically distributed random vectors in $\mathbb{R}^d$ with mean vector $\mu$ and covariance matrix $\mathbf{\Sigma}$. The Berry-Esseen theorem in the multivariate case states that the upper bound of the error between the real distribution and normalized Gaussian distribution is:*

$$\sup_{\mathbf{z} \in \mathbb{R}^d} |\mathbb{P}(\mathbf{S}_n \leq \mathbf{z}) - \Phi_{\mathbf{\Sigma}}(\mathbf{z})| \leq O(\frac{1}{\sqrt{n}}),, \tag{13}$$

*where $\mathbf{S}_n = \frac{1}{\sqrt{n}} \sum_{i=1}^n (\mathbf{X}_i - \mu)$ is the normalized sum of the random variables, $\Phi_{\mathbf{\Sigma}}(\mathbf{z})$ is the cumulative distribution function of the multivariate normal distribution with mean vector $\mathbf{0}$ and covariance matrix $\mathbf{\Sigma}$, and $\|\mathbf{X}_1 - \mu\|$ denotes the Euclidean norm.*

**Theorem 1.** *If $\sqrt{|D_t|} \gg \sum_{i=1}^t |F_i|$ and $f(\mathbf{x}; w_0) = Q_0(y|\mathbf{x})$, then $\hat{R}_t(w)$ is equivalent to $R_t(w)$.*

*Proof.* The estimated predictions of data $(\mathbf{x}, y)$ for both the remaining data and forgetting data in the $t$-th round of request is:

$$\tilde{f}_t(\mathbf{x}, w_0) = Q_t(y|\mathbf{x}) = g_t^{[y]}(\mathbf{x}) z(\mathbf{x}|y) Q_0(y|\mathbf{x}),$$

where $g_t^{[y]}(\mathbf{x}) \propto \frac{Q_t(y)}{Q_0(y)\phi_0^{[y]}(z(\mathbf{x}))}$. In the above equation, $z(\mathbf{x}|y)$ involves bias because the low-dimensional vectors cannot always fit the multivariate Gaussian distribution perfectly. Therefore, based on the Berry-Esseen theorem, we estimate the error between the real distribution and the approximate Gaussian distribution:

$$\sup_{(\mathbf{x},y)\in D_t} |z(\mathbf{x}|y) - \Phi(n(\mathbf{x}))| \leq O(\frac{1}{\sqrt{|D_t|}}),$$

where $|D_t|$ is the size of $D_t$. Therefore, for all the estimated predictions of data $(\mathbf{x}, y)$ through distribution shift, one error term exists between the estimated predictions and the optimal predictions:

$$\sup_{(\mathbf{x},y)\in D_t} |\tilde{f}_t(\mathbf{x}, w_0) - f(\mathbf{x}, w_t^*)| \leq O(\frac{1}{\sqrt{|D_t|}}).$$

Therefore, for any historical sequential unlearning requests $\{F_t\}$, the estimated risk of $\hat{R}_t(w)$ after removing $F$ can be represented by:

$$|\hat{R}_t(w) - R_t(w)| = \frac{1}{|D_t|} \sum_{i=1}^{t} \sum_{(\mathbf{x},y)\in F_i} |d_{\text{KL}}(f(\mathbf{x}; w_t), q_t(\mathbf{x})f(\mathbf{x}; w_0)) - d_{\text{KL}}(f(\mathbf{x}; w_t), f(\mathbf{x}; w_t^*))|$$

$$= \frac{1}{|D_t|} \sum_{i=1}^{t} \sum_{(\mathbf{x},y)\in F_i} (f(\mathbf{x}; w_t)|\log(q_t(\mathbf{x})f(\mathbf{x}; w_0)) - \log(f(\mathbf{x}; w_t^*)))|$$

$$\leq \frac{1}{|D_t|} \sum_{i=1}^{t} \sum_{(\mathbf{x},y)\in F_i} (f(\mathbf{x}; w_t)|\frac{1}{q_t(\mathbf{x})f(\mathbf{x}; w_0)} - \frac{1}{f(\mathbf{x}; w_t^*)})|$$

$$= \frac{1}{|D_t|} \sum_{i=1}^{t} \sum_{(\mathbf{x},y)\in F_i} (\frac{f(\mathbf{x}; w_t)}{q^t(\mathbf{x})f(\mathbf{x}; w_0)f(\mathbf{x}; w_t^*)}|q_t(\mathbf{x})f(\mathbf{x}; w_0) - f(\mathbf{x}; w_t^*))|$$

$$\leq O(\frac{\sum_{i=1}^{t} |F_i|}{\sqrt{|D_t|}}),$$

where the total size of forgetting data $\sum_{i=1}^{t} |F_i|$ is always less than the size of remaining data $|D_t|$ and $K$ is the size of remaining data and $\sqrt{|D_t|} \gg \sum_{i=1}^{t} |F_i|$, and it demonstrates that $|\hat{R}_t(w) - R_t(w)| \approx 0$ and $\hat{R}_t(w)$ is equivalent to $R_t(w)$. $\square$

**Lemma 2.** *(Ben-David et al., 2006) Let $\mathcal{R}$ be a fixed representation function from $\mathcal{X}$ to $\mathcal{Z}$ and $\mathcal{H}$ be a hypothesis space of VC-dimension $d$. If a random labeled sample of size $m$ is generated by applying $\mathcal{R}$ to a $\mathcal{D}_S$-i.i.d. sample labeled according to $f$, then with probability at least $1 - \delta$, for every $h \in \mathcal{H}$:*

$$\epsilon_T(h) \leq \hat{\epsilon}_S(h) + \sqrt{\frac{4}{m}\left(d\log\frac{2em}{d} + \log\frac{4}{\delta}\right)} + div(\mathcal{D}_S, \mathcal{D}_T)$$

*where $e$ is the base of the natural logarithm, and $div(\tilde{\mathcal{D}}_S, \tilde{\mathcal{D}}_T)$ is the distance between source domain data $\mathcal{D}_S$ and target domain data $\mathcal{D}_T$.*

The proof can be found in the proof of theorem 1 in (Ben-David et al., 2006)

**Theorem 2.** *If the risk $R_t(w)$ satisfies the upper-bounded gradient assumption with upper bound $U$, and the model weights satisfy the gradient assumption with upper bound $W$ (i.e. $|w| \leq W$, $||\nabla R_t(w)|| \leq U$). For any sequence of unlearning requests $\{F_t\}_1^T$ with the rounds to be $T$, we set*

$\gamma = \frac{\sqrt{W}}{4\sqrt{T}}$. *Then, by applying the first-order optimization algorithm:*
*(i) the error in the $t$-th rounds of unlearning compared with the optimal model state $w_t^*$ is bounded:*

$$\mathbb{E}\left[R_t(w_t) - R_t(w_t^*)\right] \leq O(\sqrt{T}).$$

*where $V_T = 1 + \sum_{t=1}^{T} ||w_t^* - w_{t-1}^*||$.*

*Proof.* Let $w_t^*$ denote the optimal model parameters in the $t$-th round of removal request. Then the difference between $w_t$ and $w_t^*$ is:

$$||w_t - w_t^*||^2 = ||w_t - w_t^* - w_{t-1}^* + w_{t-1}^*||^2$$
$$= ||w_t - w_{t-1}^*||^2 + ||w_t^* - w_{t-1}^*||^2 + 2(w_t - w_{t-1}^*)^\top(w_t^* - w_{t-1}^*). \quad (14)$$

After incorporating $w_t$ into the first item of eq 14 as shown in the following:

$$||w_t - w_{t-1}^*||^2 = ||w_{t-1} - \gamma(\frac{\nabla\hat{R}_t(w_0)}{||\nabla\hat{R}_t(w_0)||} + \frac{\nabla\hat{R}_{t-1}(w_0)}{||\nabla\hat{R}_{t-1}(w_0)||}) - w_{t-1}^*||^2$$

$$= ||w_{t-1} - w_{t-1}^*||^2 + ||\gamma(\frac{\nabla\hat{R}_t(w_0)}{||\nabla\hat{R}_t(w_0)||} - \frac{\nabla\hat{R}_{t-1}(w_0)}{||\nabla\hat{R}_{t-1}(w_0)||})||^2 -$$

$$2\gamma(\frac{\nabla\hat{R}_t(w_0)}{||\nabla\hat{R}_t(w_0)||} - \frac{\nabla\hat{R}_{t-1}(w_0)}{||\nabla\hat{R}_{t-1}(w_0)||})^\top(w_{t-1} - w_{t-1}^*). \quad (15)$$

We can incoperate Eq. 15 into Eq. 14:

$$||w_t - w_t^*||^2 \leq ||w_t^* - w_{t-1}^*||^2 + ||w_{t-1} - w_{t-1}^*||^2 + ||\gamma(\frac{\nabla\hat{R}_t(w_0)}{||\nabla\hat{R}_t(w_0)||} - \frac{\nabla\hat{R}_{t-1}(w_0)}{||\nabla\hat{R}_{t-1}(w_0)||})||^2 -$$

$$2\gamma(\frac{\nabla\hat{R}_t(w_0)}{||\nabla\hat{R}_t(w_0)||} - \frac{\nabla\hat{R}_{t-1}(w_0)}{||\nabla\hat{R}_{t-1}(w_0)||})^\top(w_{t-1} - w_{t-1}^*) +$$

$$2(w_t - w_t^*)^\top(w_t^* - w_{t-1}^*). \quad (16)$$

By rearranging terms and multiplying $\frac{1}{2\gamma}$ on both sides we have:

$$2\gamma(\frac{\nabla\hat{R}_t(w_0)}{||\nabla\hat{R}_t(w_0)||} - \frac{\nabla\hat{R}_{t-1}(w_0)}{||\nabla\hat{R}_{t-1}(w_0)||})^\top(w_{t-1} - w_{t-1}^*)$$

$$\leq \frac{1}{2\gamma}[||w_t^* - w_{t-1}^*||^2 + ||w_{t-1} - w_{t-1}^*||^2 - ||w_t - w_t^*||^2$$

$$+ ||\gamma(\frac{\nabla\hat{R}_t(w_0)}{||\nabla\hat{R}_t(w_0)||} - \frac{\nabla\hat{R}_{t-1}(w_0)}{||\nabla\hat{R}_{t-1}(w_0)||})||^2 + 2(w_t - w_{t-1}^*)^\top(w_t^* - w_{t-1}^*)]$$

$$\leq \frac{1}{2\gamma}[2W||w_t^* - w_{t-1}^*|| + ||w_{t-1} - w_{t-1}^*||^2 - ||w_t - w_t^*||^2$$

$$+ 4\gamma^2 + 4W||w_t^* - w_{t-1}^*||].$$

Therefore the estimated error of training loss $f$ in the $t$ round on the remaining data $D^t$ is:

$$\mathbb{E}_{D_t}|R_t(w_t) - R_t(w_t^*)| \leq ||U\mathbb{E}\left[w_t - w_t^*\right]||$$

$$= U\mathbb{E}\left[|\nabla(\frac{\nabla\hat{R}_{t+1}(w_0)}{||\nabla\hat{R}_{t+1}(w_0)||} - \frac{\nabla\hat{R}_t(w_0)}{||\nabla\hat{R}_t(w_0)||})^\top(w_t - w_t^*)\right]$$

$$\leq \frac{U}{2\gamma}\left[6W||w_{t+1}^* - w_t^*|| + ||w_t - w_t^*||^2 - ||w_{t+1} - w_{t+1}^*||^2 + 4\gamma^2\right]$$

$$\leq \frac{U}{2\gamma}(16W^2 + 4\gamma^2) = 32UW^{\frac{3}{2}}\sqrt{T} + 8U\frac{\sqrt{W}}{\sqrt{T}} = O(\sqrt{T}).$$

$\square$

Then, for the regularizer term, i.e. the prediction error on the forgetting data between the models after unlearning and after retraining, we have:

$$\mathbb{E}|\mathcal{R}(F_t, w_t)| = \mathbb{E}|\mathcal{L}(F_t, w_t)) - \mathcal{L}(F_t, w_t^*)) - \mathcal{L}(D_t, w_t)) + \mathcal{L}(D_t, w_t^*) + \mathcal{L}(D_t, w_t) - \mathcal{L}(D_t, w_t^*)|$$
$$\leq \mathbb{E}|\mathcal{L}(F_t, w_t)) - \mathcal{L}(D_t, w_t))| + \mathbb{E}|\mathcal{L}(F_t, w_t^*)) - \mathcal{L}(D_t, w_t^*))| + \mathbb{E}|\mathcal{L}(D_t, w_t) - \mathcal{L}(D_t, w_t^*)|$$

$$\leq \mathbb{E}|R_{D_t}^t(w_t) - R_{D_t}^t(w_t^*)| + O(\sqrt{\frac{\log |D_t|}{|D_t|}}) + 2\mathrm{div}(D_t, F_t)$$

$$\leq O(\sqrt{T}) + 2\mathrm{div}(D_t, F_t)$$

Combining the estimated errors in the training loss and the regularizer, we can get an error bound of the online unlearning risk in the $t$ round:

$$\mathbb{E}|R^t(w_t) - R^t(w_t^*)| \leq O(\sqrt{T}),$$

where $\mathrm{div}(D_t, F_t)$ denotes the divergence between the distribution of remaining data $D^t$ and $\mathrm{div}(D_t, F_t) \in (0, 1)$.

**Theorem 3.** *If the risk $R_t(w)$ satisfies the upper-bounded gradient assumption with upper bound $U$, and the model weights satisfy the gradient assumption with upper bound $W$ (i.e. $|w| \leq W$ $||\nabla R_t(w)|| \leq U$). For any sequence of unlearning requests $\{F_t\}_1^T$ with the rounds to be $T$, we set $\gamma = \frac{\sqrt{W}}{4\sqrt{T}}$. Then, by applying the first-order optimization algorithm:*
*(ii) the accumulated unlearning regret across all requests is bounded:*

$$\mathbb{E}\left[U\text{-}Regret_T(\{w_t\}_1^T)\right] \leq O(V_T\sqrt{T} + \Delta_T),$$

*where $V_T = 1 + \sum_{t=1}^T ||w_t^* - w_{t-1}^*||$ and $\Delta_T = 2\sum_{t=1}^T div(D_t, F_t)$ and $div(D_t, F_t)$ denotes the divergence between the distribution of remaining data $D_t$ and forgetting data $F_t$ in the $t$-th round of unlearning.*

*Proof.* We can get the upper bound of the regret estimation by summing the upper bound of $\mathcal{L}(D_t, w_t) - \mathcal{L}(D_t, w_t^*)$ for each $t = 1, \ldots, K$:

$$|\mathbb{E}\sum_{t=1}^T (\mathcal{L}(D_t, w_t) - \mathcal{L}(D_t, w_t^*))|$$

$$= \sum_{t=0}^{T-1} |\mathbb{E}(\mathcal{L}(D_t, w_t) - \mathcal{L}(D_t, w_t^*))| + |\mathbb{E}(\mathcal{L}(D_T, w_T) - \mathcal{L}(D_T, w_T^*))| - |\mathbb{E}(\mathcal{L}(D_0, w_0) - \mathcal{L}(D_0, w_0^*))|$$

$$\leq \sum_{t=0}^{T-1} U\mathbb{E}[|\nabla(\frac{\nabla \hat{R}_{t+1}(w_0)}{||\nabla \hat{R}_{t+1}(w_0)||} - \frac{\nabla \hat{R}_t(w_0)}{||\nabla \hat{R}_t(w_0)||})^\top (w_t - w_t^*)] + O(\sqrt{T})$$

$$\leq U \sum_{t=0}^{T-1} \frac{1}{2\gamma} \left[||w_{t+1}^* - w_t^*|| + ||w_t - w_t^*||^2 - ||w_{t+1} - w_{t+1}^*||^2 + 4\gamma^2\right] + O(\sqrt{T})$$

$$\leq \frac{U}{2\gamma} \left[||w_0 - w_0^*||^2 - ||w_T - w_T^*||^2 + 4T\gamma^2 + \sum_{t=0}^{T-1} 6W||w_{t+1}^* - w_t^*||\right] + O(\sqrt{T})$$

$$\leq \frac{U}{2\gamma} \left[4W^2 + 4T\gamma^2 + 6W(V_T - 1)\right] + O(\sqrt{T})$$

$$= \frac{16UW^{\frac{3}{2}} + UW^{\frac{1}{2}} + 12UW^{\frac{1}{2}}(V_T - 1)}{2}\sqrt{T} + O(\sqrt{T})$$

$$= O(V_T\sqrt{T}), \tag{17}$$

where $V_T = 1 + \sum_{t=1}^T ||w_t^* - w_{t-1}^*||$.

$$
\begin{aligned}
\mathbb{E}\left[U\text{-}Regret_T\right] \leq & \left|\mathbb{E}\sum_{t=1}^{T}(\mathcal{L}(D_t, w_t) - \mathcal{L}(D_t, w_t^*))\right| + \mathbb{E}\sum_{t=1}^{T}\mathcal{R}(F_t, w_t) \\
\leq & 2\left|\mathbb{E}\sum_{t=1}^{T}(\mathcal{L}(D_t, w_t) - \mathcal{L}(D_t, w_t^*))\right|2\sum_{t=1}^{T}\mathrm{div}(D_t, F_t) \\
= & O(V_T\sqrt{T}) + 2\sum_{t=1}^{T}\mathrm{div}(D_t, F_t).
\end{aligned} \tag{18}
$$

$\square$

# B EXPERIMENTS

In this section, we provide a detailed description of the datasets, models, baseline methods, and the implementation details of the online unlearning algorithm. We then present further experimental results to answer the following six research questions, which are crucial for evaluating the online unlearning algorithm:

- **RQ1**: How does the performance of SAFE compare to other methods in terms of accuracy, JS-Divergence, and Membership Inference Attack?

- **RQ2**: How does online unlearning perform under different settings of rounds and forgetting data size compared to other methods?

- **RQ3**: How does the hyperparameter, specifically the learning rate, affect the unlearning performance?

- **RQ4**: What impact do the distribution shift risk, the population risk of forgetting data, and the gradient of the original model have on the unlearning performance?

- **RQ5**: How does SAFE perform in terms of memory computation efficiency?

Each of these questions is addressed through comprehensive experimental analyses to thoroughly evaluate the capabilities and limitations of the SAFE algorithm.

## B.1 DATASETS AND MODELS

In the experiments, we choose four datasets: **DIGITS** (MNIST) (LeCun, 1998), **FASHION** (Fashion MNIST) (Xiao et al., 2017), **CIFAR10 Features**, and raw **CIFAR10** (Krizhevsky et al., 2009). We use the original DIGITS, FASHION, and CIFAR10 datasets from *torchvision module*. For CIFAR10 Features, we choose a well-trained **ResNet-18** model to extract the 512-dimensional features.

For the experiment models, we choose the **CNN**(LeCun et al., 1995) with two convolutional layers for the two MNIST datasets. The output channels for the two convolutional layers are 16 and 32, respectively. Then, the other parts of the CNN consist of three linear layers with output dimensions of 256, 128, and 10. Then for the CIFAR10 Feature dataset, we choose a binary linear regression model with an input size of 512. Finally, for the CIFAR10 datasets, we choose an **ResNet-18** (He et al., 2016) with two linear layers with the output dimensions 256 and 10 and the **ResNet** does not contain the pre-trained weights.

## B.2 BASELINES

We compare the performance of the SAFE algorithm with **Retrain** which are the standard results from the retraining models, and four other state-of-the-art unlearning works with high efficiency and potential to handle sequential unlearning requests: L-CODEC (Mehta et al., 2022), Descent-U (Neel et al., 2021), Unrolling (Thudi et al., 2022), and CaMU (Shen et al., 2024a). **L-CODEC** (Mehta et al., 2022) first apply the pruning strategy to select the model parameters that are associated with the selected forgetting data and then apply gradient ascent algorithm as shown in (Sekhari et al., 2021) on the selected parameters for unlearning; **Descent-U** (Neel et al., 2021) calculate the gradients of remaining data and use the perturbed gradient descent for unlearning; **Unroll** (Thudi et al., 2022) records gradients when learning the first epoch and adds recorded gradients on weights after the incremental training; **CaMU** construct the counterfactual samples for each forgetting sample and implement unlearning on both representation and prediction levels.

Table 4: Complete comparison results in 20 rounds of unlearning, which remove 400 data points (avg%±std%). The **bold** record indicates the best, and the underlined record indicates the second-best.

| Method | RA | UA | TA | Remain JS | Forget JS | Test JS | ASR |
|---|---|---|---|---|---|---|---|
| | | | | **MNIST** | | | |
| Retrain | 99.68±0.05 | 98.89±0.09 | 99.00±0.05 | 0.00±0.00 | 0.00±0.00 | 0.00±0.00 | 79.25±1.14 |
| Unroll (Thudi et al., 2022) | 99.24±0.22 | **98.91±0.15** | 98.61±0.19 | 0.59±0.11 | 0.88±0.11 | 0.89±0.10 | 79.27±1.15 |
| CaMU (Shen et al., 2024a) | 98.94±0.36 | 98.72±0.79 | 98.54±0.42 | 12.12±2.29 | 13.43±1.56 | 11.98±2.22 | 79.05±1.13 |
| LCODEC (Mehta et al., 2022) | 96.26±1.95 | 96.27±1.88 | 95.60±1.94 | 2.61±1.13 | 2.87±1.25 | 2.93±1.25 | 78.79±2.22 |
| Descent-U (Neel et al., 2021) | 98.78±0.53 | 98.72±0.53 | 98.27±0.46 | 0.89±0.32 | 1.05±0.31 | 1.13±0.28 | 79.24±1.14 |
| SAFE | **99.74±0.03** | 99.63±0.05 | **99.04±0.02** | **0.34±0.03** | **0.61±0.06** | **0.65±0.03** | **79.26±1.15** |
| | | | | **MNIST Fashion** | | | |
| Retrain | 96.44±0.15 | 90.76±0.33 | 90.40±0.15 | 0.00±0.00 | 0.00±0.00 | 0.00±0.00 | 79.57±0.51 |
| Unroll (Thudi et al., 2022) | 90.61±0.91 | 89.08±0.94 | 87.99±0.85 | 4.21±0.52 | 4.75±0.56 | 4.82±0.53 | **79.14±0.65** |
| CaMU (Shen et al., 2024a) | 91.32±0.36 | 90.45±0.83 | 89.00±0.40 | 10.40±0.45 | 11.53±0.43 | 10.51±0.41 | 78.07±0.53 |
| LCODEC (Mehta et al., 2022) | 86.21±4.27 | 86.26±4.15 | 82.28±3.89 | 7.81±2.57 | 8.60±2.38 | 8.94±2.45 | 77.41±1.75 |
| Descent-U (Neel et al., 2021) | 89.52±1.38 | 89.27±1.54 | 87.40±1.10 | 4.81±0.75 | 4.99±0.73 | 5.19±0.67 | 78.28±0.77 |
| SAFE | **91.73±0.25** | **90.60±0.43** | **89.04±0.30** | **3.62±0.15** | **4.09±0.25** | **4.20±0.17** | 77.96±0.49 |
| | | | | **CIFAR10 Feature** | | | |
| Retrain | 85.61±0.13 | 84.16±0.57 | 85.01±0.16 | 0.00±0.00 | 0.00±0.00 | 0.00±0.00 | 48.72±1.95 |
| Unrolling (Thudi et al., 2022) | 86.25±0.06 | **84.97±0.27** | 85.51±0.03 | 0.06±0.02 | 0.07±0.02 | 0.06±0.02 | 49.32±1.92 |
| CaMU (Shen et al., 2024a) | 85.77±0.12 | **83.40±0.62** | **84.99±0.15** | 0.12±0.03 | 0.12±0.03 | 0.13±0.03 | 51.50±1.62 |
| LCODEC (Mehta et al., 2022) | 85.98±0.06 | 84.91±0.27 | 85.34±0.03 | **0.05±0.02** | **0.06±0.02** | **0.06±0.02** | **48.82±1.88** |
| Descent-U (Neel et al., 2021) | 86.24±0.07 | 84.94±0.30 | 85.50±0.01 | 0.07±0.02 | 0.07±0.02 | 0.07±0.02 | 49.45±1.91 |
| SAFE | **85.74±0.13** | 84.10±1.15 | 84.94±0.14 | 0.13±0.03 | 0.14±0.04 | 0.13±0.04 | 49.38±2.38 |
| | | | | **CIFAR10** | | | |
| Retrain | 97.61±0.25 | 91.78±0.49 | 91.19±0.34 | 0.00±0.00 | 0.00±0.00 | 0.00±0.00 | 64.38±1.34 |
| Unroll (Thudi et al., 2022) | 93.59±2.57 | 90.46±1.92 | 86.97±2.12 | 4.02±1.38 | 5.96±1.09 | 5.82±1.27 | 74.96±2.95 |
| CaMU (Shen et al., 2024a) | **95.71±1.09** | 93.13±3.32 | **89.58±1.09** | 4.96±0.82 | 7.38±2.95 | 6.14±0.85 | 74.54±2.95 |
| LCODEC (Mehta et al., 2022) | 23.95±3.71 | 24.08±4.18 | 23.33±3.54 | 49.21±2.62 | 48.93±8.42 | 48.81±2.85 | 54.47±8.42 |
| Descent-U (Neel et al., 2021) | 74.01±27.50 | 74.24±27.58 | 70.01±25.34 | 17.63±17.06 | 17.66±16.6 | 17.66±16.4 | **71.95±7.67** |
| SAFE | 94.22±0.85 | **92.30±0.45** | 87.74±0.84 | **3.46±0.46** | **5.67±0.28** | **5.37±0.28** | 74.18±3.31 |

## B.3 IMPLEMENTATION DETAILS

All the experiments are conducted on one server with NVIDIA RTX A5000 GPUs (24GB GDDR6 Memory) and 12th Gen Intel Core i7-12700K CPUs (12 cores and 128GB Memory). The code of SAFE was implemented in Python 3.9.16 and Cuda 11.6.1. The main Python packages' versions are the following: Numpy 1.23.5; Pandas 2.0.1; Pytorch 1.13.1; Torchvision 0.14.1.

All the experiments on these baselines are conducted under 10 random seeds based on the original models trained in the four datasets. We train two CNN models on two MNIST datasets for 20 epochs with a learning rate of 1e-3 and a weight decay of 1e-4. In contrast, we train the logistic regression model on the CIFAR10 Feature dataset for 30 epochs using an Adam optimizer with a learning rate 0.05. We train another ResNet-18 model on the CIFAR10 dataset for 20 epochs, where the learning rate is set as 0.1 and other hyperparameters are the same as the code[2]. For the two MNIST datasets, the batch size is set as 32, and for the other two datasets, the batch size is 128. For the retrained models, we adopt the same hyperparameters as the training process of the original model.

In real implementations, to save the GPU memory cost, we divide the risk estimator $\hat{R}_t$ calculation into two phases. First, we calculate and record the accumulated gradient $R^0(w_0)$ on all training data, and store all the gradients in a backup model. Then, we calculate the distribution shift loss and population risk for forgetting data and record the gradient on the second backup model. Then we add the gradients of each parameter together because the different parts in $\hat{R}_t$ are linearly added.

Then, the hyperparameters used in the implementation of SAFE only include the amplification factor of the distribution shift loss and the learning rate. For the amplification factor, we set it as 1000 for two MNIST datasets and the CIFAR10 Feature dataset, and we set it as 120000 for the CIFAR10

---

[2]https://github.com/kuangliu/pytorch-cifar/tree/master

dataset. Then for the two MNIST datasets and the CIFAR10 Feature dataset, which uses models with simpler structures, we set the learning rate as $\frac{\sqrt{W}}{4\sqrt{T}}$, where $W$ is the maximal parameters of the original model and $T$ is the total unlearning rounds. Then for the CIFAR10 dataset, we set the learning rate as $\frac{5\sqrt{W}}{3\sqrt{T}}$.

For the evaluations, we assess the unlearning algorithm using five metrics: **RA**, **UA**, and **TA**, which denote the prediction accuracy of the post-unlearning model on the remaining data, forgetting data, and test data. The closer value to the retrained model indicates better unlearning performance for these metrics. We also compare **JS** of the instance predictions of the post-unlearning and retrained models to evaluate the instance-level performance better. Lower **JS** stands for the better results. We also check the **ASR**, the attack accuracy of the MIA (Shokri et al., 2017; Chen et al., 2021). Specifically, we choose the same MIA evaluation as (Fan et al., 2024; Jia et al., 2023). Specifically, we use the subset of remaining data with the size of 10000 as positive data and real test data with the size of 10000 as negative data to construct the attacker model's training set. Then, we train an SCV with the Radial Basis Function Kernel model as the attacker. Then, the attacker was evaluated using the forgetting data to measure attack success rates.

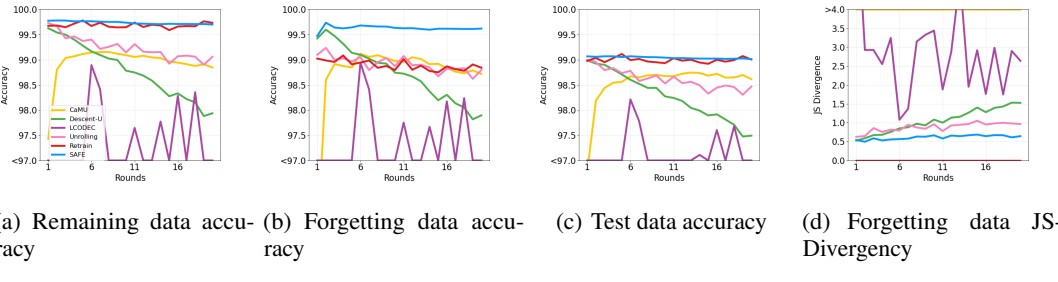

(a) Remaining data accuracy  (b) Forgetting data accuracy  (c) Test data accuracy  (d) Forgetting data JS-Divergency

Figure 3: Model performance against unlearning rounds on MNIST.

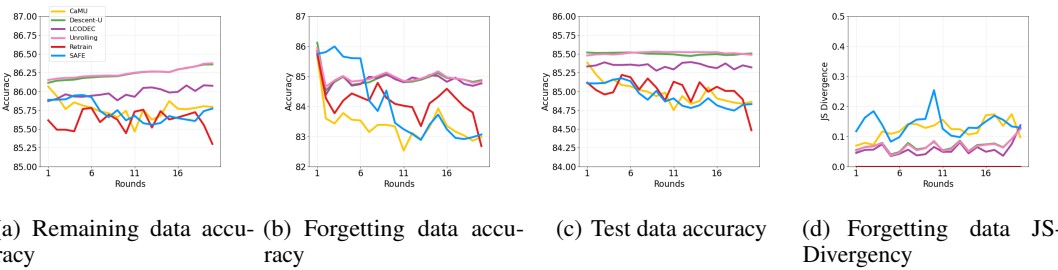

(a) Remaining data accuracy  (b) Forgetting data accuracy  (c) Test data accuracy  (d) Forgetting data JS-Divergency

Figure 4: Model performance against unlearning rounds on CIFAR10 Feature.

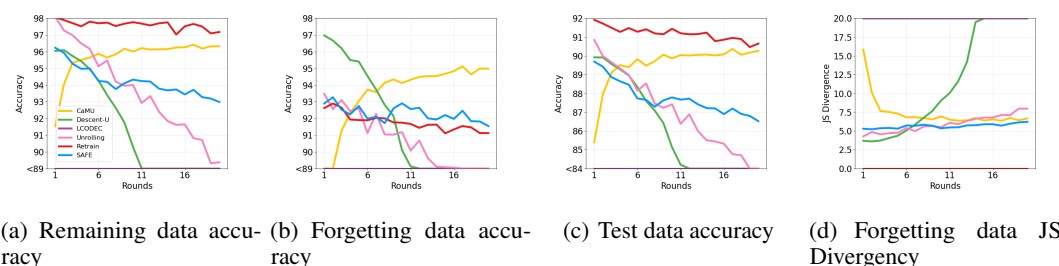

(a) Remaining data accuracy  (b) Forgetting data accuracy  (c) Test data accuracy  (d) Forgetting data JS-Divergency

Figure 5: Model performance against unlearning rounds on CIFAR10.

### B.4 COMPLETE PERFORMANCE COMPARISON

First, we present the complete experimental results in Table 4, and we illustrate the online unlearning process for the MNIST, CIFAR10 Feature, and CIFAR10 datasets in Figure 3, Figure 4, and Figure 5, respectively.

In Figure 3, the results for SAFE are depicted with blue lines, while the results for the retrained model are shown with red lines. Figure 3(a) and Figure 3(b) demonstrate that the remaining data accuracy and forgetting data accuracy of SAFE are the closest to those of the retrained model compared with other methods. Additionally, in Figure 3(c), SAFE maintains high prediction performance in all 20 rounds. Figure 3(d) further demonstrates the effectiveness of removing forgetting data information, as both the group-level accuracy and instance-level divergence indicate that SAFE achieves results closest to those of the retrained model on forgetting data. In Figure 4(a) and Figure 4(b), the remaining data accuracy and forgetting data accuracy of SAFE continue to be the closest to those of the retrained model compared with other methods. Furthermore, in Figure 4(c), SAFE maintains the closest test accuracy to the retrained model. However, in Figure 4(d), the divergence in forgetting data accuracy is higher than in other methods, which can be mainly attributed to the larger adjustments made to the model by SAFE. Although Figure 5(a) and Figure 5(c) show that SAFE can lead to performance degradation in remaining data and test data accuracies, Figure 5(b) and Figure 5(d) still demonstrate that SAFE achieves results closest to the retrained model on forgetting data, both in terms of group-level accuracy and instance-level divergence.

Table 5: Effect analysis on size and rounds of forgetting requests on MNIST (avg%±std%).

| Method | RA | UA | TA | JS | GAP | RA | UA | TA | JS | GAP |
|---|---|---|---|---|---|---|---|---|---|---|
| | 400 Samples for 10 Rounds | | | | | 800 Samples for 10 Rounds | | | | |
| Retrain | 99.69±0.04 | 98.94±0.08 | 98.99±0.05 | 0.00±0.00 | 0.00 | 99.70±0.03 | 98.78±0.05 | 98.99±0.05 | 0.00±0.00 | 0.00 |
| Unroll | 99.80±0.04 | 99.60±0.05 | 99.08±0.03 | **0.58±0.05** | 0.36 | 99.76±0.03 | 99.51±0.09 | 99.04±0.03 | **0.67±0.03** | 0.38 |
| CaMU | 98.90±0.52 | **98.50±1.12** | 98.38±0.57 | 12.31±1.30 | 3.54 | 98.94±0.15 | **98.55±0.35** | 98.55±0.21 | 14.25±1.20 | 3.92 |
| LCODEC | 29.34±5.34 | 29.31±5.13 | 29.54±5.36 | 43.85±2.87 | 63.32 | 27.82±3.64 | 27.90±3.23 | 28.02±3.72 | 44.56±1.80 | 64.57 |
| Descent-U | **99.64±0.04** | 99.62±0.04 | **98.98±0.04** | 0.60±0.06 | 0.33 | 99.64±0.04 | 99.59±0.04 | 98.98±0.04 | 0.69±0.03 | 0.39 |
| SAFE | 99.76±0.02 | 99.59±0.07 | 99.07±0.02 | 0.59±0.05 | 0.35 | **99.68±0.02** | 99.44±0.10 | **98.99±0.02** | 0.73±0.03 | 0.35 |
| | 400 Samples for 40 Rounds | | | | | 800 Samples for 20 Rounds | | | | |
| Retrain | 99.69±0.04 | 98.84±0.09 | 98.97±0.06 | 0.00±0.00 | 0.00 | 99.69±0.03 | 98.79±0.05 | 98.97±0.06 | 0.00±0.00 | 0.00 |
| Unroll | 98.78±0.05 | **98.72±0.04** | 98.27±0.04 | 1.05±0.06 | 0.70 | 99.74±0.04 | 99.54±0.07 | 99.01±0.03 | **0.67±0.03** | 0.38 |
| CaMU | 98.72±0.36 | 98.50±0.66 | 98.47±0.32 | 15.75±2.72 | 4.39 | 98.69±0.33 | **98.34±0.37** | 98.44±0.24 | 16.29±2.40 | 4.57 |
| LCODEC | 28.78±4.24 | 28.84±4.15 | 28.92±4.27 | 44.27±2.45 | 63.81 | 29.49±4.48 | 29.98±4.51 | 29.77±4.50 | 43.59±2.29 | 62.95 |
| Descent-U | 99.40±0.18 | 99.37±0.18 | 98.79±0.15 | 0.76±0.12 | 0.44 | 99.56±0.09 | 99.53±0.07 | 98.92±0.07 | 0.72±0.05 | 0.41 |
| SAFE | **99.72±0.02** | 99.67±0.03 | **99.03±0.02** | **0.64±0.05** | 0.39 | **99.69±0.03** | 99.55±0.06 | **99.00±0.03** | 0.70±0.03 | 0.37 |

Table 6: Effect analysis on size and rounds of forgetting requests on MNIST Fashion (avg%±std%).

| Method | RA | UA | TA | JS | GAP | RA | UA | TA | JS | GAP |
|---|---|---|---|---|---|---|---|---|---|---|
| | 400 Samples for 10 Rounds | | | | | 800 Samples for 10 Rounds | | | | |
| Retrain | 96.40±0.18 | 90.82±0.44 | 90.48±0.14 | 0.00±0.00 | 0.00 | 96.43±0.15 | 91.15±0.53 | 90.36±0.14 | 0.00±0.00 | 0.00 |
| Unroll | 90.92±0.68 | 89.05±0.94 | 88.29±0.60 | 4.56±0.50 | 3.50 | 90.68±0.64 | 88.95±0.96 | 88.03±0.61 | 7.94±0.51 | 4.56 |
| CaMU | 91.44±0.34 | **90.39±0.84** | 88.99±0.44 | 11.82±0.49 | 4.68 | 91.30±0.25 | 90.63±5.27 | 89.03±2.23 | 11.83±0.56 | 4.70 |
| LCODEC | 25.65±4.35 | 25.47±4.79 | 25.38±4.29 | 44.18±3.16 | 61.34 | 24.49±3.08 | 24.50±3.80 | 24.14±2.99 | 45.63±2.46 | 62.61 |
| Descent-U | **93.06±0.30** | 92.97±0.49 | **90.17±0.18** | 3.29±0.20 | 2.27 | **93.05±0.30** | 93.33±0.63 | **90.17±0.18** | 3.22±0.27 | 2.24 |
| SAFE | 93.02±0.10 | 92.18±0.30 | 90.13±0.09 | 3.42±0.20 | 2.13 | 91.96±0.48 | **90.67±0.90** | 89.14±0.49 | 4.10±0.43 | 2.57 |
| | 400 Samples for 40 Rounds | | | | | 800 Samples for 20 Rounds | | | | |
| Retrain | 96.56±0.19 | 90.71±0.25 | 90.24±0.21 | 0.00±0.00 | 0.00 | 96.53±0.20 | 90.80±0.53 | 90.21±0.21 | 0.00±0.00 | 0.00 |
| Unroll | 90.54±0.81 | 89.31±0.96 | 87.89±0.74 | 4.73±0.47 | 3.62 | 90.79±0.72 | 89.28±0.94 | 88.08±0.68 | 4.63±0.48 | 3.50 |
| CaMU | 90.75±0.71 | 89.95±0.77 | 88.67±0.52 | 11.68±0.48 | 4.95 | 90.75±0.61 | **89.97±0.86** | 88.66±0.44 | 11.95±0.57 | 5.03 |
| LCODEC | 24.79±4.13 | 25.33±4.41 | 24.57±4.06 | 44.74±2.44 | 61.89 | 25.13±4.52 | 25.18±4.83 | 24.78±4.43 | 44.86±3.10 | 61.83 |
| Descent-U | 92.22±0.60 | 92.12±0.60 | **89.60±0.42** | **3.66±2.62** | 2.51 | **92.69±0.46** | 92.75±0.76 | **89.93±0.32** | **3.46±0.33** | 2.38 |
| SAFE | **92.30±0.67** | **91.50±0.63** | 89.43±0.61 | 3.81±0.37 | 2.42 | 92.54±0.37 | 91.66±0.60 | 89.61±0.36 | 3.77±0.31 | 2.30 |

Table 7: Effect analysis on size and rounds of forgetting requests on CIFAR10 Feature (avg%±std%).

| Method | RA | UA | TA | JS | GAP | RA | UA | TA | JS | GAP |
|---|---|---|---|---|---|---|---|---|---|---|
| | **400 Samples for 10 Rounds** | | | | | **800 Samples for 10 Rounds** | | | | |
| Retrain | 84.69±0.65 | 83.28±1.43 | 84.34±0.70 | 0.00±0.00 | 0.00 | 84.45±0.92 | 83.06±1.87 | 84.04±1.08 | 0.00±0.00 | 0.00 |
| Unroll | 86.20±0.02 | 85.02±0.33 | 85.51±0.02 | 0.17±0.15 | 1.15 | 86.22±0.04 | 85.45±0.19 | 85.50±0.01 | 0.22±0.20 | 1.46 |
| CaMU | 85.76±0.13 | **83.43±0.57** | 85.06±0.14 | 0.20±0.13 | 0.54 | 85.81±0.10 | 84.31±0.52 | 85.07±0.13 | 0.24±0.18 | 0.97 |
| LCODEC | 85.93±0.03 | 84.95±0.36 | 85.34±0.03 | **0.14±0.15** | 1.01 | 85.97±0.05 | 85.38±0.22 | 85.35±0.03 | **0.20±0.21** | 1.34 |
| Descent-U | 86.18±0.03 | 84.97±0.41 | 85.51±0.75 | 0.17±0.15 | 1.13 | 86.19±0.05 | 85.46±0.20 | 85.51±0.75 | 0.22±0.20 | 1.46 |
| SAFE | **85.23±0.15** | 84.67±1.24 | **84.75±0.20** | 0.30±0.14 | 0.66 | **85.16±0.10** | **83.55±0.90** | **84.57±0.20** | 0.33±0.16 | 0.51 |
| | **400 Samples for 40 Rounds** | | | | | **800 Samples for 20 Rounds** | | | | |
| Retrain | 83.09±2.27 | 79.28±5.02 | 82.03±3.09 | 0.00±0.00 | 0.00 | 84.44±0.85 | 82.89±1.66 | 83.91±1.05 | 0.00±0.00 | 0.00 |
| Unroll | 86.38±0.15 | 84.96±0.19 | 85.40±0.04 | 0.61±0.60 | 3.24 | 86.29±0.09 | 85.43±0.15 | 85.49±0.02 | 0.23±0.19 | 1.55 |
| CaMU | **85.81±0.11** | **83.09±0.59** | **84.83±0.22** | **0.57±0.50** | 2.47 | 85.82±0.11 | 84.17±0.37 | 84.97±0.13 | 0.25±0.17 | 0.99 |
| LCODEC | 86.09±0.14 | 84.91±0.20 | 85.34±0.04 | 0.59±0.61 | 3.13 | 86.03±0.08 | 85.34±0.17 | 85.35±0.03 | **0.20±0.19** | 1.42 |
| Descent-U | 86.36±0.15 | 84.98±0.22 | 85.50±0.01 | 0.61±0.60 | 3.26 | 86.27±0.09 | 85.44±0.15 | 85.50±0.01 | 0.23±0.19 | 1.55 |
| SAFE | 86.07±0.10 | 84.03±0.71 | 85.05±0.17 | 0.59±0.54 | 2.83 | **85.76±0.09** | **84.13±0.53** | **84.89±0.12** | 0.27±0.17 | 0.95 |

Table 8: Effect analysis on size and rounds of forgetting requests on CIFAR10 (avg%±std%).

| Method | RA | UA | TA | JS | GAP | RA | UA | TA | JS | GAP |
|---|---|---|---|---|---|---|---|---|---|---|
| | **400 Samples for 10 Rounds** | | | | | **800 Samples for 10 Rounds** | | | | |
| Retrain | 97.75±0.15 | 92.15±0.39 | 91.44±0.22 | 0.00±0.00 | 0.00 | 97.57±0.25 | 91.48±0.50 | 91.11±0.41 | 0.00±0.00 | 0.00 |
| LCODEC | 24.17±4.37 | 24.43±4.97 | 23.55±4.17 | 48.69±3.21 | 64.47 | 22.06±1.48 | 22.16±1.62 | 22.51±1.44 | 50.19±1.82 | 65.91 |
| Unroll | **95.97±1.23** | **92.32±0.67** | 88.95±1.04 | **4.94±0.41** | 2.34 | 92.21±2.71 | 87.88±1.87 | 85.75±2.28 | 7.02±1.12 | 5.34 |
| CaMU | 95.10±1.31 | 91.46±4.34 | **89.02±1.30** | 8.27±2.72 | 3.51 | 95.62±0.48 | 93.16±1.72 | 89.52±0.46 | 7.57±0.76 | 3.20 |
| LCODEC | 24.17±4.37 | 24.43±4.97 | 23.55±4.17 | 48.69±3.21 | 64.47 | 22.06±1.48 | 22.16±1.62 | 22.51±1.44 | 50.19±1.82 | 65.91 |
| Descent-U | 94.07±1.89 | 94.42±2.09 | 88.23±1.64 | 5.39±1.80 | 3.64 | 94.04±1.92 | **93.00±2.12** | 88.19±1.58 | 5.52±1.81 | 3.37 |
| SAFE | 91.21±1.68 | 86.62±1.09 | 84.98±1.58 | 8.43±0.48 | 6.74 | **96.10±0.17** | 94.57±0.64 | **89.52±0.20** | **4.76±0.18** | 2.73 |
| | **400 Samples for 40 Rounds** | | | | | **800 Samples for 20 Rounds** | | | | |
| Retrain | 97.10±0.64 | 90.97±0.96 | 90.45±0.87 | 0.00±0.00 | 0.00 | 97.06±0.68 | 90.65±1.03 | 90.35±0.93 | 0.00±0.00 | 0.00 |
| LCODEC | 23.55±3.19 | 23.47±3.53 | 22.90±3.04 | 49.16±2.44 | 64.44 | 22.70±2.11 | 22.72±1.98 | 22.12±1.95 | 49.54±1.77 | 65.02 |
| Unroll | 89.34±4.90 | 86.84±4.16 | 83.37±4.16 | 8.06±2.39 | 6.76 | 87.77±5.07 | 84.29±4.05 | 81.87±4.41 | 9.12±2.38 | 8.31 |
| CaMU | **96.03±0.84** | 93.95±2.68 | **89.84±0.82** | 7.35±1.56 | 3.00 | 95.73±0.43 | **93.63±1.37** | **89.56±0.44** | 7.01±0.80 | 3.03 |
| LCODEC | 23.55±3.19 | 23.47±3.53 | 22.90±3.04 | 49.16±2.44 | 64.44 | 22.70±2.11 | 22.72±1.98 | 22.12±1.95 | 49.54±1.77 | 65.02 |
| Descent-U | 90.75±9.54 | **90.95±9.59** | 85.43±8.40 | 6.85±5.18 | 4.56 | 72.98±28.9 | 72.86±28.8 | 69.00±26.7 | 18.06±17.0 | 20.32 |
| SAFE | 95.88±0.52 | 95.52±0.31 | 89.33±0.55 | **4.85±0.54** | 2.94 | **95.84±0.15** | 94.47±0.57 | 89.14±0.23 | **5.17±0.39** | 2.85 |

## B.5 FURTHER COMPARISON UNDER DIFFERENT SETTINGS

In the following four tables: Table 5, Table 6, Table 7, and Table 8, we present additional experimental results under different unlearning request settings. For each dataset, we conduct four groups of experiments with varying configurations:

- Setting the unlearning round to 10 and removing 400 samples in each round.
- Setting the unlearning round to 40 and removing 400 samples in each round.
- Setting the unlearning round to 10 and removing 800 samples in each round.
- Setting the unlearning round to 20 and removing 800 samples in each round.

In these extended experiments, SAFE consistently achieves better results compared to other methods, confirming the findings of our previous experiments.

## B.6 EFFECT ANALYSIS OF LEARNING RATE

In this part, we show experiments on the hyperparameter tuning, where we choose different learning rates $\gamma = \frac{\sqrt{W}}{K\sqrt{T}}$ in the online update algorithm. On the MNIST, MNIST-Fashion, and CIFAR10 Feature datasets, we choose $K = 2, 4, 6, 8, 10, 12$ to conduct experiments on the corresponding $\gamma$, and on CIFAR10 dataset, we choose $K = 0.2, 0.4, 0.6, 0.8, 1.0, 1.2$ as conduct experiments on the corresponding $\gamma$. The plots of model performances in different rounds is shown in Figure 6, Figure 7, Figure 8, and Figure 9.

In the first three figures, we can see the lower K, which implies the higher $\gamma$, can usually result in more significant changes in the model parameters and then lead to larger degradation on the remaining data, forgetting data, and test data accuracies while the lower K usually results in smaller drops on all the three type of accuracies because it only makes minor differences on the model. Then, after considering the balance of remaining data accuracy and forgetting data accuracy, we choose $K = 4$ on the MNIST, MNIST-Fashion, and CIFAR10 Feature datasets. When $K = 4$, the remaining data accuracies are quite close to other results, and when $K > 4$, the forgetting data accuracies are much lower. On the CIFAR10 dataset, we set $K = 0.6$, which can also result in reaching a better balance between remaining data accuracy and forgetting data accuracy.

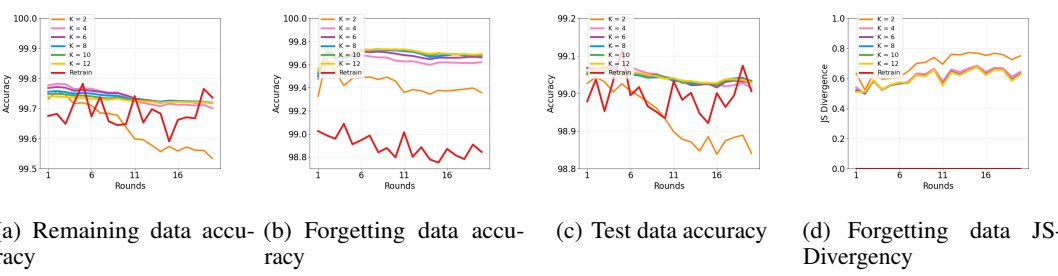

(a) Remaining data accuracy  (b) Forgetting data accuracy  (c) Test data accuracy  (d) Forgetting data JS-Divergency

Figure 6: Model performance against unlearning rounds on MNIST.

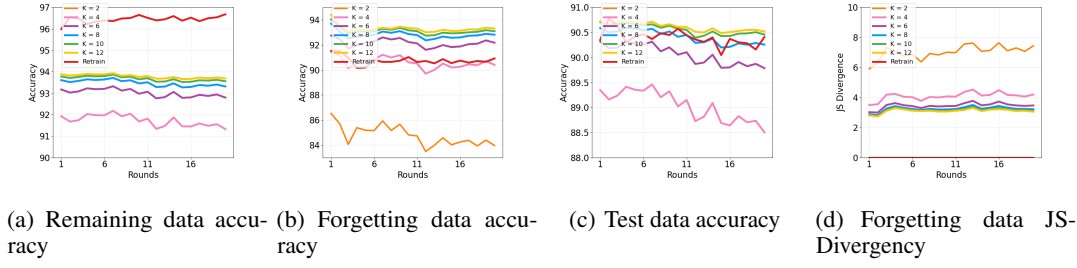

(a) Remaining data accuracy  (b) Forgetting data accuracy  (c) Test data accuracy  (d) Forgetting data JS-Divergency

Figure 7: Model performance against unlearning rounds on MNIST Fashion.

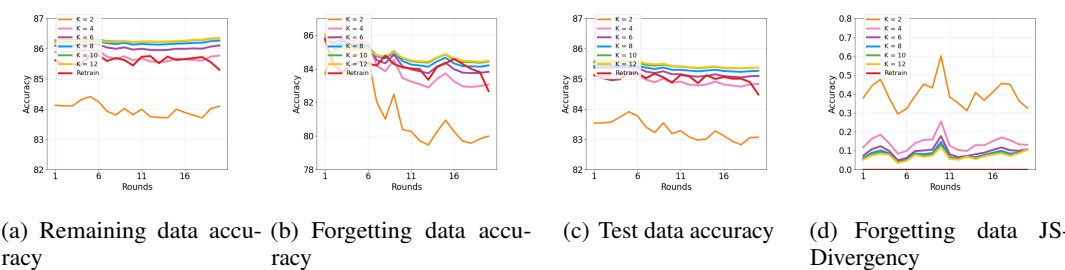

(a) Remaining data accuracy  (b) Forgetting data accuracy  (c) Test data accuracy  (d) Forgetting data JS-Divergency

Figure 8: Model performance against unlearning rounds on CIFAR10 Feature.

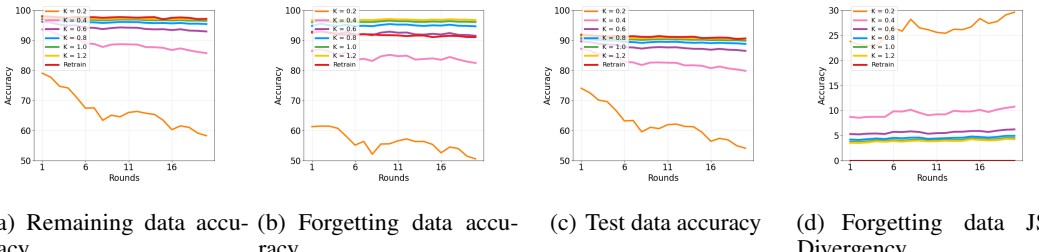

(a) Remaining data accuracy  (b) Forgetting data accuracy  (c) Test data accuracy  (d) Forgetting data JS-Divergency

Figure 9: Model performance against unlearning rounds on CIFAR10.

Table 9: Abaltion study results (avg%±std%).

| Method | RA | UA | TA | Remain JS | Forget JS | Test JS | ASR |
|---|---|---|---|---|---|---|---|
| | | | | **MNIST** | | | |
| Retrain | 99.68±0.05 | 98.89±0.09 | 99.00±0.05 | 0.00±0.00 | 0.00±0.00 | 0.00±0.00 | 79.25±1.14 |
| w/o DSR | 99.81±0.01 | 99.78±0.02 | 99.06±0.01 | **0.30±0.03** | **0.57±0.05** | **0.62±0.02** | 79.27±1.15 |
| w/o FR | 99.75±0.02 | 99.65±0.05 | 99.05±0.02 | 0.33±0.03 | 0.61±0.05 | 0.65±0.03 | **79.26±1.15** |
| w/o TR | 99.54±0.01 | **99.39±0.07** | 98.88±0.01 | 0.45±0.03 | 0.71±0.04 | 0.75±0.03 | 79.26±1.15 |
| **SAFE** | **99.74±0.03** | 99.63±0.05 | **99.04±0.02** | 0.32±0.03 | 0.61±0.04 | 0.65±0.03 | 79.26±1.15 |
| | | | | **MNIST Fashion** | | | |
| Retrain | 96.44±0.15 | 90.76±0.33 | 90.40±0.15 | 0.00±0.00 | 0.00±0.00 | 0.00±0.00 | 79.57±0.51 |
| w/o DSR | **92.23±0.01** | 92.00±0.32 | **89.58±0.01** | **3.35±0.08** | **3.65±0.16** | **3.91±0.09** | 77.52±0.51 |
| w/o FR | 91.85±0.21 | **90.80±0.38** | 89.16±0.24 | 3.56±0.13 | 4.00±0.22 | 4.14±0.14 | 77.89±0.57 |
| w/o TR | 89.38±0.32 | 87.47±0.60 | 86.86±0.30 | 4.77±0.17 | 5.48±0.19 | 5.36±0.19 | **79.17±0.53** |
| **SAFE** | 91.73±0.25 | 90.60±0.43 | 89.04±0.30 | 3.62±0.15 | 4.09±0.25 | 4.20±0.17 | 77.96±0.49 |
| | | | | **CIFAR10 Feature** | | | |
| Retrain | 85.61±0.13 | 84.16±0.57 | 85.01±0.16 | 0.00±0.00 | 0.00±0.00 | 0.00±0.00 | 48.72±1.95 |
| w/o DSR | 85.89±0.03 | 85.87±0.41 | 85.10±0.03 | 0.13±0.04 | 0.15±0.05 | 0.13±0.04 | 46.91±1.94 |
| w/o FR | 85.88±0.03 | 85.38±0.82 | **85.07±0.08** | **0.12±0.03** | **0.13±0.03** | **0.12±0.03** | 47.56±2.15 |
| w/o TR | 85.31±1.08 | 82.34±3.24 | 84.68±7.40 | 0.18±0.03 | 0.18±0.03 | 0.18±0.03 | 52.42±1.63 |
| **SAFE** | **85.74±0.13** | 84.10±1.15 | 84.94±0.14 | 0.13±0.03 | 0.14±0.04 | 0.13±0.04 | **49.38±2.38** |
| | | | | **CIFAR10** | | | |
| Retrain | 97.61±0.25 | 91.78±0.49 | 91.19±0.34 | 0.00±0.00 | 0.00±0.00 | 0.00±0.00 | 64.38±1.34 |
| w/o DSR | **97.35±0.01** | 97.44±0.10 | **90.84±0.01** | **2.06±0.07** | **3.96±0.19** | **3.75±0.09** | **74.10±3.33** |
| w/o FR | 94.42±0.76 | 92.59±0.39 | 87.92±0.76 | 3.36±0.42 | 5.55±0.24 | 5.30±0.51 | 74.18±3.31 |
| w/o TR | 92.30±1.05 | 89.36±0.66 | 86.03±1.04 | 4.36±0.60 | 6.85±0.38 | 6.36±0.69 | 74.18±3.18 |
| **SAFE** | 94.22±0.85 | **92.30±0.45** | 87.74±0.84 | 3.46±0.46 | 5.67±0.28 | 5.37±0.28 | 74.18±3.31 |

## B.7 ABLATION STUDY

Table 9 presents the results of the ablation study for the proposed algorithm, where we sequentially remove the distribution shift loss (**DSR**), the forgetting data gradient (**FR**), and the initial training data gradient (**TR**).

First, when the distribution shift loss is removed, the forgetting data accuracies on all four datasets are similar to the original forgetting data accuracies, implying that the information of the streaming forgetting data has not been effectively removed from the model. Second, when only the forgetting data gradient is removed, the forgetting data accuracies approach those of the retrained models. However, the results of the complete SAFE algorithm still outperform those without the forgetting data gradient. Lastly, when the initial training data gradient is removed, there is a significant drop in forgetting data accuracies, remaining data accuracies, and test accuracies across all datasets.

These experimental results indicate that the distribution shift loss and the forgetting data gradient contribute significantly to the unlearning process. The distribution shift loss is the dominant factor, while the forgetting data gradient also provides a substantial contribution. Additionally, the initial training data gradient is crucial in maintaining overall performance.

In addition, from the comparison of JS-Divergences, we observe that although experiments without distribution shift loss and forgetting data gradient do not achieve perfect unlearning, they still result in lower JS-Divergences compared to the retrained model. This phenomenon indicates that minor adjustments to the original model can lead to lower JS-Divergences on all the remaining data, as well as forgetting and test data. This implies that the unlearning algorithm can produce prediction results similar to those of the retrained model. However, there remain differences in the model parameters and soft prediction results compared to the retrained models.

## B.8 MEMORY EFFICIENCY ANALYSIS

Figure 10 presents the comparison results of the maximal memory cost on both CPU and GPU during unlearning. Although SAFE can reach the highest time efficiency, it does not require significant memory cost as LCODEC (Mehta et al., 2022). On the two MNIST and CIFAR10_Feature datasets, it only requires 20% extra GPU memory and 6% extra CPU memory than retraining during unlearning. On the CIFAR10 dataset, SAFE requires 50% extra CPU memory, which ranks the medium position among all methods and 40% extra GPU memory. Considering the significant improvement in time efficiency, the insignificant increase in the memory requirement still shows the superiority of SAFE on the algorithm efficiency.

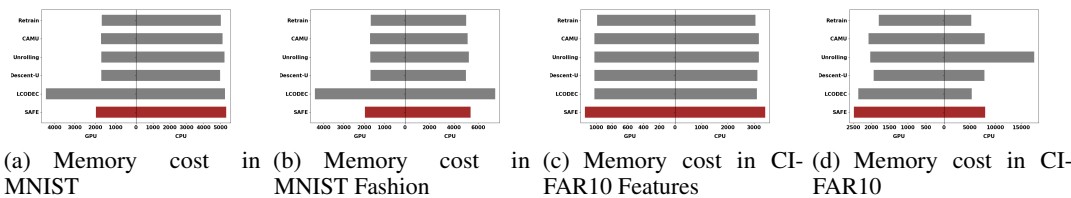

(a) Memory cost in MNIST (b) Memory cost in MNIST Fashion (c) Memory cost in CIFAR10 Features (d) Memory cost in CIFAR10

Figure 10: Maximal memory cost comparisons.

## B.9 EXPERIMENTS ON LARGE DATASET

We also conduct experiments on a larger dataset, TinyImagenet, and we present the results in Table 10. Compared to LCODEC and Descent-U, which do not require access to training data during unlearning, SAFE significantly outperforms these methods in both effectiveness and efficiency. Importantly, SAFE achieves the fastest computation time at just 7.30 seconds, demonstrating its high efficiency. When compared to Unroll and CaMU (which require the original training data), SAFE's performance remains competitive while being much more efficient. Overall, SAFE strikes an excellent balance between accuracy and efficiency, particularly when training data is unavailable.

Table 10: Complete comparison results in 20 rounds of unlearning, which remove 400 data points.

| Method | RA | UA | TA | Remain JS | Forget JS | ASR | Time |
|---|---|---|---|---|---|---|---|
| | | | | TinyImagenet | | | |
| Retrain | 75.57±0.51 | 42.71±0.57 | 42.50±0.54 | 0.00±0.00 | 0.00±0.00 | 29.69±3.31 | 762.00 |
| Unroll (Thudi et al., 2022) | 53.87±2.00 | 48.62±1.99 | 42.08±1.42 | 21.68±0.98 | 23.93±0.82 | 24.20±0.83 | 22.72 |
| CaMU (Shen et al., 2024a) | 38.60±2.14 | 36.17±3.78 | 35.06±1.98 | 41.94±4.11 | 40.16±3.27 | 39.07±3.87 | 16.21 |
| LCODEC (Mehta et al., 2022) | 9.34±5.19 | 9.17±5.21 | 6.84±3.29 | 54.67±5.81 | 53.88±3.62 | 54.67±3.62 | 13.04 |
| Descent-U (Neel et al., 2021) | 27.50±11.18 | 27.48±11.30 | 24.63±9.65 | 38.52±7.55 | 37.56±6.71 | 37.73±6.75 | 20.70 |
| SAFE | **48.69±5.28** | **46.35±4.05** | **36.28±3.97** | **25.80±1.41** | **28.06±1.03** | **33.42±2.17** | **7.30** |

## C GAUSSIAN DISTRIBUTION VERIFICATION

Before unlearning, we standardize the low-dimensional vectors such that their mean vector is zero and the covariance matrix is the identity matrix. As each unlearning round progresses, we update the mean vector and covariance matrix. Although the low-dimensional vectors continue to follow a Gaussian distribution, the exact distributions may not be identical across rounds because we removed the vectors of different data in different classes. To verify the Gaussian nature of the

vector distribution, we employ Mardia's test, which is highly effective in examining multivariate Gaussian distributions Mardia (1970). The table below presents the Skewness and Kurtosis p-values of Mardia's test [1] on the low-dimensional vectors for each class throughout the unlearning process. The consistently high Skewness and Kurtosis p-values suggest that the low-dimensional vectors maintain a Gaussian distribution across all unlearning rounds.

| Dataset | Mnist | Fashion | Cifar_feature | Cifar |
|---|---|---|---|---|
| Skewness p_value | 1.00 | 1.00 | 1.00 | 1.00 |
| Kurtosis p_value | 0.93 | 0.95 | 0.87 | 0.84 |

Table 11: P-values of skewness and kurtosis tests for different datasets

## C.1 STREAM SETTING FOR SINGLE CLASS UNLEARNING

To further evaluate the effectiveness of the SAFE algorithm on single-class data unlearning tasks, we designed experiments that combine stream unlearning with class-wise unlearning. In each round, the unlearning request involves removing a subset of data belonging to the same class, continuing until the entire class is removed. We refer to this process as stream-for-class unlearning. We conducted experiments on the MNIST, CIFAR10, and TinyImagenet datasets to assess the feasibility of stream-for-class unlearning. For the TinyImagenet dataset, we choose the ResNet-18 model as well. In these experiments, we primarily evaluate four metrics: **RA** (training accuracy on the remaining classes), **UA** (training accuracy on the forgotten class), **TA(R)** (test accuracy on the remaining classes), and **TA(U)** (test accuracy on the forgetting class). Similar to our previous experiments, we performed 20 rounds of unlearning, where, by the 20th round, all data from the target class is removed, transitioning the task to full-class unlearning.

Among the five baseline methods, Retrain, Unroll, and CaMU perform single-batch unlearning on all accumulated data in each request, while LCODEC, Descent-U, and SAFE apply stream unlearning for each request. Table 12 shows the results after completing the final round of unlearning. On the TinyImagenet and CIFAR10 datasets, SAFE achieves the best average performance across the four metrics, demonstrating the most balanced trade-off between target class unlearning and preserving knowledge of other classes. Although SAFE does not outperform the batch unlearning methods on the two MNIST datasets, it still delivers close and comparable results even if it achieves the unlearning through a stream manner.

Table 12: Last round performance comparisons for stream unlearning on a single class(avg%±std%).

| Method | RA | UA | TA(R) | TA(U) | GAP | RA | UA | TA(R) | TA(U) | GAP |
|---|---|---|---|---|---|---|---|---|---|---|
| | | | **MNIST** | | | | | **MNIST-Fashion** | | |
| Retrain | 99.71±0.18 | 0.00±0.00 | 99.01±0.21 | 0.00±0.00 | 0.00 | 97.36±0.39 | 0.00±0.00 | 92.49±0.35 | 0.00±0.00 | 0.00 |
| LCODEC | 27.92±2.38 | 26.42±3.13 | 28.04±2.37 | 26.02±3.11 | 44.24 | 24.40±23.03 | 13.11±17.89 | 24.17±17.63 | 13.13±22.49 | 82.93 |
| Unroll | 97.36±0.33 | 0.00±0.00 | 97.70±0.28 | **0.00±0.00** | 0.73 | 87.30±4.86 | 0.00±0.00 | 85.61±4.72 | **0.00±0.00** | 3.39 |
| CaMU | **98.90±0.13** | 0.00±0.00 | **98.69±0.14** | 0.00±0.00 | 0.23 | 91.68±0.92 | 0.00±0.00 | 89.91±1.00 | **0.00±0.00** | 1.65 |
| Descent-U | 21.87±7.34 | 6.72±7.47 | 22.56±7.99 | 6.64±7.93 | 34.86 | 20.20±19.9 | 8.97±4.05 | 20.28±19.0 | 8.70±4.02 | 35.15 |
| SAFE | 96.53±0.51 | 1.19±0.24 | 95.96±0.46 | 0.95±0.10 | 1.86 | **93.66±0.16** | 7.86±0.48 | **93.02±0.43** | 6.34±1.00 | 4.95 |
| | | | **TinyImageNet** | | | | | **CIFAR10** | | |
| Retrain | 65.05±0.74 | 0.00±0.00 | 42.90±0.53 | 0.00±0.00 | 0.00 | 94.38±0.25 | 0.00±0.00 | 86.63±0.35 | 0.00±0.00 | 0.00 |
| LCODEC | 8.76±13.00 | 0.55±0.18 | 7.71±11.36 | 0.53±0.17 | 18.62 | 62.09±43.14 | 21.29±29.85 | 61.59±42.91 | 20.55±27.95 | 23.94 |
| Unroll | 35.36±2.51 | **0.00±0.00** | 27.39±1.60 | **0.00±0.00** | 9.04 | 87.05±2.21 | **0.00±0.00** | 82.34±1.84 | **0.00±0.00** | 2.32 |
| CaMU | 38.96±1.67 | 9.08±4.45 | 28.81±1.13 | 8.14±3.58 | 13.11 | 94.04±0.49 | 0.11±0.09 | 89.49±0.56 | 0.23±0.23 | 0.75 |
| Descent-U | 1.29±0.19 | **0.00±0.00** | 1.19±0.23 | **0.00±0.00** | 21.09 | 55.04±7.96 | 7.03±8.30 | 53.49±7.86 | 6.97±8.20 | 18.69 |
| SAFE | **52.88±0.10** | 5.40±0.15 | **36.39±0.06** | 7.21±0.21 | 7.70 | **94.30±0.17** | 0.29±0.08 | **88.75±0.15** | 0.10±0.04 | 0.54 |

# D LIMITATIONS AND FUTURE WORK

As the first work to introduce online scenarios and dynamic regret into the unlearning problem, this paper has some limitations.

First, in the theoretical part, we prove an upper bound of dynamic regret proportional to $(1 + V_T)\sqrt{T}$. However, this upper bound is suboptimal because we cannot determine the real value of $V_T$ in experiments, making it difficult to obtain a tighter upper bound. Second, in the experimental part, we

use the maximal weight of the original model as the upper bound of the model weight $W$. However, the weight of the updated model may exceed $W$, which could affect the accuracy of our results. Third, in practical applications, the size of the forgetting data at each time point of the streaming unlearning requests may not be fixed. For experimental convenience, we fixed the size of the forgetting data in this work, but this does not fully reflect real-world scenarios.

These limitations inspire several directions for future work. First, we aim to find better optimization approaches and more effective learning rates to narrow the upper bound of dynamic regret, ensuring both theoretical correctness and practical feasibility. Second, we plan to explore more extensions of this work in practical unlearning applications, particularly focusing on irregular streaming unlearning requests and the sample-to-class unlearning task.

