# OpenReview forum: "Machine Unlearning for Streaming Forgetting"
_ICLR.cc/2025/Conference — Submitted to ICLR 2025_

### Official Review · Reviewer_3ki7 · 2024-10-31

**Soundness:** 2
**Presentation:** 1
**Contribution:** 2
**Rating:** 3
**Confidence:** 3

**Summary:**

This paper addresses the challenge of machine unlearning, a process where users can request that their data be removed from a model’s knowledge in compliance with recent regulatory requirements. Unlike traditional batch-based approaches to unlearning, which handle all deletion requests in a single step, this work focuses on the streaming scenario where requests arrive sequentially. The authors frame the problem through an online learning lens, where each unlearning request is treated as a distribution shift to efficiently remove data incrementally. They propose a novel online unlearning algorithm that does not require access to the original training data, ensuring that knowledge derived from forgotten data is removed while preserving insights from the remaining data. The paper's theoretical analysis provides guarantees on error bounds by quantifying the cumulative regret based on total variation and divergence shifts between data to be forgotten and retained. The method achieves guaranteed regret bounds without assuming convexity of the loss function. Experimental results across various datasets and models demonstrate the method's practicality and efficiency in real-world streaming unlearning scenarios.

**Strengths:**

* The online learning perspective is novel, and in retrospective makes a lot of sense for sequential unlearning requests.
* The algorithm is practical and seems on par with baselines according to the experimental results.

**Weaknesses:**

* **Heuristic Approach to Unlearning.** The paper analyzes one heuristic approach to unlearning (Problem (1)), which jointly minimizes the loss on the retain and the KL divergence (from the model obtained via retraining from scratch) on the forget data. Unfortunately, it is not clear that this heuristic attains the objective of unlearning, which is *being statistically indistinguishable from retraining without the forget data*. There are rigorous definitions of the unlearning objective (Ginart et al. 2019, Guo et al. 2020, Sekhari et al. 2021) which quantify the level of statistical indistinguishability and which are algorithmically attainable, and several works analyzing sequential (adaptive) requests towards this objective (Gupta et al. 2021, Chourasia et al. 2023). It is a missed opportunity to provide theoretical guarantees on the regret of the algorithm, without any guarantees on the central issue at hand, which is unlearning.

* **Insufficient Experimental Validation of Unlearning.** While there is no theoretical unlearning guarantee as mentioned above, there is not enough experimental evidence on unlearning neither. This is typically done through membership inference attacks, which are deferred to the appendix where they are not discussed enough, nor is the specific membership inference attack explained. The latter point is quite important, since it is not trivial to adapt membership inference attacks for machine unlearning (Hayes et al., 2024).

* **Meaningfulness of the Regret Bound.** It is not clear when the regret bound is meaningful, even assuming that the additive term (sum of divergence of retain datasets from forget datasets) is negligible. The term $V_T$ is the sum of distance between successive optimal parameters, and one would expect each summand to be in the order of $\Omega_T(1)$, which would make the final regret bound superlinear and thus vacuous.

* **Clarity of Theoretical and Algorithmic Components.** The paper is quite unclear/hard to read at several occurrences:

1. For example, Theorem 1 claims that "$\hat{R}_t(w)$ is equivalent to $R_t(w)$" but the meaning of the so-called equivalence is not given.
2.  Moreover, the KL divergence in line 111 is not well-defined a priori since it is not clear that the problem is a classification task and whether the arguments should be the distribution over classes or the parameter/model distributions.
3. Line 7 of Algorithm 1 is unclear: "Calculate the shift distribution risk and population risk for forgetting data"; it is not clear what is exactly is the shift distribution risk nor how to calculate it.
4. $div(D_t, F_t)$ appears in Theorem 2 and Lemma 2 without being defined; "the divergence" is not a precise description.
5. Lines 357-359 on the comparison with the results of Sekhari et al. (2021) are not justified and unclear.
6. In the experiments setting, the "JS" metric is not explained and assumed to be known by the reader.

## References
- Ginart et al. (NeurIPS 2019). Making ai forget you: Data deletion in machine learning.
- Guo et al. (ICML 2020). Certified data removal from machine learning models.
- Sekhari et al. (NeurIPS 2021). Remember what you want to forget: Algorithms for machine unlearning.
- Gupta et al. (NeurIPS 2021). Adaptive machine unlearning.
- Chourasia et al. (ICML 2023). Forget unlearning: Towards true data-deletion in machine learning.
- Hayes et al. (2024). Inexact Unlearning Needs More Careful Evaluations to Avoid a False Sense of Privacy.

**Questions:**

in addition to addressing the concerns above, here are some questions and suggestions:
* Can the term $V_T$ be analyzed for a simple convex problem? This would help clarify the practical relevance of the regret bound.

* Could the authors expand on the membership inference attack used in experiments? Which specific attack was employed, and were any adjustments made to suit the unlearning setting?

* Future work might benefit from exploring advanced membership inference techniques or theoretical unlearning guarantees, even if initially focused on convex problems.

---

> ### Author Response · Authors · 2024-11-24
> **Response to Reviewer 3ki7**
>
> Thank you for the reviews. Below we provide responses to the weaknesses raised as well as address the questions.
>
> **W1 Heuristic Approach to Unlearning. The paper analyzes one heuristic approach to unlearning (Problem (1)), which jointly minimizes the loss on the retain and the KL divergence (from the model obtained via retraining from scratch) on the forget data. Unfortunately, it is not clear that this heuristic attains the objective of unlearning, which is being statistically indistinguishable from retraining without forgetting data. There are rigorous definitions of the unlearning objective (Ginart et al. 2019, Guo et al. 2020, Sekhari et al. 2021) which quantify the level of statistical indistinguishability and which are algorithmically attainable, and several works analyzing sequential (adaptive) requests towards this objective (Gupta et al. 2021, Chourasia et al. 2023). It is a missed opportunity to provide theoretical guarantees on the regret of the algorithm, without any guarantees on the central issue at hand, which is unlearning.**
>
> The main argument raised in the Weakness 1 is that our unlearning objective Eq. 1, is purely heuristic and unrelated to formal unlearning definitions. We respectfully disagree with this argument. From an optimization perspective, the optimal model obtained via retraining without forgetting data is also the optimal solution to Eq. 1 under appropriate conditions. In other words, if Eq. 1 is successfully optimized, the resulting model is not only statistically indistinguishable from retraining without the forget data—as required by formal unlearning definitions—but is mathematically equal to the retraining model when $\lambda=0$, and all theoretical guarantees also apply to this $\lambda=0$ scenario.
>
> The inclusion of the KL divergence penalty term on the forgetting data serves an additional purpose: it ensures that the unlearned model can have a similar performance on the forgetting data as the retrained model. This introduces an important degree of flexibility, allowing control over the performance of forgetting data, a consideration that is absent in prior works. This flexibility may be crucial in practical applications where some performance requirement for forgetting data is explicitly desirable.
>
> Overall, our approach aligns with the rigorous definitions proposed by (Ginart et al. 2019, Guo et al. 2020, and Sekhari et al. 2021) while extending their frameworks to incorporate additional practical considerations, specifically on forgetting data.
>
> **W2 Insufficient Experimental Validation of Unlearning. While there is no theoretical unlearning guarantee, as mentioned above, there is not enough experimental evidence on unlearning either. This is typically done through membership inference attacks, which are deferred to the appendix where they are not discussed enough, nor is the specific membership inference attack explained. The latter point is quite important since it is not trivial to adapt membership inference attacks for machine unlearning (Hayes et al., 2024).** and **Q2 Could the authors expand on the membership inference attack used in experiments? Which specific attack was employed, and were any adjustments made to suit the unlearning setting?**
>
> Thank you for raising these important concerns about our handling of membership inference attacks (MIA) in the paper. We have addressed your feedback in the updated version as follows:
>
> - We have moved the MIA analysis from the appendix to the main text.
> - We expanded the details of the MIA methodology. Specifically, we adapted the approach in [1] and [2], using the remaining data (positive) and real test data (negative) as the attacker model’s training set. The attacker was then evaluated on the forgetting data to measure attack success rates, providing insights into how well the unlearning method prevents data memorization. The implementation details are described in Lines 1145–1150 of the updated paper.
> - We extended the discussion of MIA results to analyze not only which unlearning methods perform better but also why they do so. More factors such as the nature of the forgetting mechanism are explored in detail (Lines 437–443).
>
> We hope these updates address the concerns raised and improve the clarity of our experimental validation.

---

> > ### Author Response · Authors · 2024-11-24
> > **Response to Reviewer 3ki7**
> >
> > **W3: Meaningfulness of the Regret Bound. It is not clear when the regret bound is meaningful, even assuming that the additive term (sum of divergence of remaining datasets from forgetting datasets) is negligible. The term $V_T$ is the sum of the distance between successive optimal parameters, and one would expect each summand to be in the order of $\Omega_T(1)$, which would make the final regret bound superlinear and thus vacuous.** and **Q1: Can the term $V_T$ be analyzed for a simple convex problem? This would help clarify the practical relevance of the regret bound.**
> >
> >
> > Thank you for raising this important question regarding the analysis of $V_T$. To address this, we provide an analysis inspired by the insights in [3], which provides a systematic analysis of $V_T$ in online settings and adapts them to the streaming unlearning problem.
> >
> > 1. Behavior of $V_T$
> > - Constant $O(1)$: For tasks with minimal parameter variation over time (e.g., static or nearly stationary problems), $V_T$ remains constant.
> > - Sublinear to $T$ ($O(T^\gamma)$, $\gamma<1$): For tasks where gradients grow smoothly as $T$ increases (e.g., problems with diminishing changes in the loss landscape over time), $V_T$ grows sublinearly.
> > - Linear to $T$ ($O(T)$): For tasks where parameter variation remains stable and unlearning requests introduce consistent changes, $V_T$ scales linearly with $T$.
> > 2. Impact on the Regret Bound
> > - Based on these scenarios, the regret bound ranges from $O(\sqrt{T})$ to $O(T^{3/2})$, depending on the specific properties of the task and unlearning dynamics.
> > 3. Comparison with Prior Work
> > - Even in the worst-case scenario $O(T^{3/2})$, the regret bound remains meaningful in comparison to existing work. For example, [4] studies the batch unlearning setting and provides an error bound that scales quadratically with the unlearning set size. When extended to streaming unlearning, the unlearning set size typically grows linearly with $T$, resulting in a regret bound of $O(T^2)$. Thus, our regret bound offers a $O(\sqrt{T)$ improvement even in the most challenging scenarios, showing its meaningfulness.
> >
> >
> > **W4: Clarity of Theoretical and Algorithmic Components. The paper is quite unclear/hard to read on several occurrences:**
> >
> > We appreciate your valuable feedback regarding the clarity of the theoretical and algorithmic components. In response, we have made significant revisions to the main text to enhance readability and precision. Below are the updates and clarifications:
> >
> > - **Theorem 1 - Equivalence Explanation**: The term *equivalence* in Theorem 1 refers to the fact that the optimizer of  $\hat{R}_t(w)$ is the optimizer for $R_t(w)$, and vice versa.
> > - **KL Divergence Clarification**: The KL divergence measures the divergence between probability distributions. Specifically, it refers to the divergence between the distributions of the model's outputs after unlearning and the estimated prediction distributions.
> > - **Shift Distribution Risk**: The shift distribution risk is defined as $d_{\text{KL}}(f(\mathbf{x};w), q_t(\mathbf{x})f(\mathbf{x}; w_0))$. The detailed calculation and its context are provided in Section 3.3 Line 264 to 272
> > - **Definition of \( \text{div}(D_t, F_t) \) in Theorem 2 and Lemma 2**: For $\text{div}(D_t, F_t)$, we adopt the definition provided in [5]. It quantifies the divergence between the source marginal $D_t$ and the target marginal $F_t$. Specifically, we use the $L_1$ or variational distance:
> >      $$
> >      \text{div}(D_t, F_t) = 2 \sup_{B \in \mathcal{B}} \lvert P_{D_t}[B] - P_{F_t}[B] \rvert,
> >      $$
> >      where $\mathcal{B}$ is the set of measurable subsets under $D_t$ and $F_t$.
> > - **Comparison with Sekhari et al. (2021)**: We have updated Lines 356–360 to provide a detailed comparison with the results of Sekhari et al. (2021). Specifically, their work shows that the error bounds are proportional to the square of the forgetting data size $O(m^2)$. In the streaming setting, as the forgetting set size $m$ grows linearly with the number of unlearning requests $T$, the resulting error bound scales as $O(T^2)$.
> > - **Jensen–Shannon Divergence (JS)**: JS, originally introduced in Line 382–386, stands for the Jensen–Shannon divergence. We have updated the text to explicitly state that it measures the divergence between the outputs of the model after unlearning and the model retrained on the remaining data. A lower JS indicates smaller differences between the two models and, therefore, better performance of the unlearning method.

---

> > > ### Author Response · Authors · 2024-11-24
> > > **Response to Reviewer 3ki7**
> > >
> > > **Q3: Future work might benefit from exploring advanced membership inference techniques or theoretical unlearning guarantees, even if initially focused on convex problems.**
> > >
> > > Thank you for providing potential future directions. Our current analysis focuses on non-convex and non-Lipschitz problems. In future work, we plan to explore ways to relax these conditions to strengthen our analysis under newly developed techniques.
> > >
> > >
> > > [1] Fan, et al. "SalUn: Empowering Machine Unlearning via Gradient-Based Weight Saliency in Both Image Classification and Generation." ICLR, 2024.
> > >
> > > [2] Liu, et al. "Model sparsity can simplify machine unlearning." NeurIPS, 2024.
> > >
> > > [3] Besbes, et al. Non-stationary stochastic optimization. Operations research, 63(5), 1227-1244, 2015.
> > >
> > > [4] Sekhari, et al. "Remember what you want to forget: Algorithms for machine unlearning." NeurIPS, 2021.
> > >
> > > [5] Ben-David, Shai, et al. "Analysis of representations for domain adaptation." NIPS, 20065

---

> ### Comment · Reviewer_3ki7 · 2024-11-25
>
> - The authors have not addressed my main concern regarding guarantees on unlearning. I understand that Problem (1) could have a nice interpretation in relation with unlearning, but it still seems that there is no rigorous link with established definitions of unlearning, i.e., statistical indistinguishability from retraining from scratch (see references in my original review).
> - If the authors concede to the point above, then perhaps the positioning of the paper should be rather empirical--which is also fine. However, regarding the empirical guarantees for unlearning which basically is success of membership inference attacks, it seems that the reported attack efficacies in Table 1 are almost all in the same range! There is unfortunately no interpretation of this part of the result, which is also a weakness of the current version of the paper. Could the author explain how their proposed method compares with existing ones in terms of defending against membership inference attacks?
> - Thanks for expanding on the regret bound further. It seems that there are cases where the regret bound is superlinear. Do the authors agree that the bound is therefore vacuous in that case? Furthermore, are there *practical cases* (precise conditions) where this is guaranteed not happen, i.e., regret bound is surely sublinear?
> - Thanks for adding a detailed comparison with Sekhari et al. Can the authors confirm that in the aforementioned work, there is an omitted dependence on the size of the dataset $n>m$, that yields a final bound of the form $m^2/n^2$? If so, this result does not appear weaker than the potential superlinear regret bound.

---

> > ### Author Response · Authors · 2024-12-02
> >
> > Thank you for your continued feedback.
> >
> > Regarding the link between the definition of this paper and the statistical indistinguishability from the retrained model, the bounded gradient assumption in Theorem 2 of our work satisfies the requirements of **Theorem 2 in [FR1]**. Consequently, it can be proven that our proposed unlearning method constitutes an $(\epsilon, \delta)$-certified removal and shows statistical indistinguishability from the retrained model. We will include the relevant proof to establish this connection in our revised version.
> >
> > Then, for the second concern about MIA, the MIA results of the MNIST dataset in Table are kept in the same range because the MNIST dataset is simple, and both the training and test predictions all have high confidence. Therefore, the attack performances of the attacker model are not satisfied. Then, for some more complex datasets, like FashionMnist, CIFAR10 and TinyImagenet, the best attack successful rates have significant differences with the second-best ones under 95% confidence. In addition, compared with other baselines, our proposed method can achieve the significantly best MIA performance on TinyImagenet and, secondly, the best MIA performance on CIFAR10. Then, although it only achieves the medium rank on FashionMnist, the results of our method have no statistical differences from the second-best results from Descent. The results prove that our method can achieve comparable or better performance with other baselines under MIA evaluations.
> >
> > Thirdly, regarding the regret bound, we agree that the superlinear regret bound will lead the online learning algorithm to fail to predict when $V_T$ is upper bounded by $O(T)$. However, in the sequential unlearning problem, the $V_T$ is more likely to be $O(1)$ or $O(T^{\gamma}), \gamma < 1$. In the stream unlearning problem, $V_T$ will be upper bound by $O(T)$ if the training loss continuously changes for all online learning rounds. This means that the predictions of the optimized model on the part of the training will always change continuously, and the decision boundaries of the model will change continuously because of the removed data points. This cannot happen for all the training data, especially for the data points with high prediction confidence. Removing such data will not have much influence on the model predictions and decision boundaries. This can also be proved by the curves of remaining data accuracies of the retrained model in Figures 1(a), 3(a), 4(a), and 5(a). Therefore, during the stream unlearning, the model predictions on most of the training data usually remain similar, and therefore, $V_T$ will not always change with the increase of $T$, and $V_T$ is more likely to be $O(1)$ or $O(T^{\gamma})$.
> >
> > Finally, regarding the bound in Sekhari et al. in the stream setting, the size of original training data $n$ will be kept as a fixed value, and only the size of forgetting data $m$ will change linearly to $T$. Therefore, the final error in Sekhari et al will be bounded by $O(m^2/n^2) = O(m^2) = O(T^2)$.

---

> > > ### Comment · Reviewer_3ki7 · 2024-12-03
> > >
> > > Thanks for your reply. I will keep my score for the moment, and discuss more with other reviewers.
> > >
> > > Regarding the connection to certified unlearning: I unfortunately do not have the reference [FR1] that the authors refer to. In any case, it would be hard to judge of the validity of this connection at this stage without further review.
> > >
> > > Regarding the MIA score: it is not clear that the proposed method performs better than others in more complex datasets, and I believe the values reported to still be very close, and the difference to not be statistically significant overall.

---

### Official Review · Reviewer_3cnv · 2024-11-02

**Soundness:** 3
**Presentation:** 2
**Contribution:** 3
**Rating:** 5
**Confidence:** 4

**Summary:**

The paper proposes a technique for online data unlearning. Thorough mathematical analysis shows that using this method under mild assumptions provides competitive regret bound guarantees. Some empirical evidence is presented to support theoretical results.

**Strengths:**

S1. The streaming setup is interesting and a novel way to think about the problem that seems useful. I think this paper would be interesting to both the “unlearning” community as well as the online learning community.

S2. The theoretical derivations have clear assumptions that do not seem excessive. Bounded gradients (Lipschitz) is a mild assumption in general. The theoretical derivations are correct, at least from my checking. (I do have some minor concerns listed later.)

S3. Some evidence for the practical effectiveness versus other methods is demonstrated in the experimental results.

**Weaknesses:**

W1. While the theoretical grounding is excellent, the experiments are too small scale to be convincing and the statistical analysis is poor or missing. The use of only small benchmarks raises doubts as to whether the relative improvement to baselines is generalizable beyond the small image datasets. Can the authors explain the choice of datasets? Why not use other datasets that are larger (such as ImageNet or variations thereof)? Why focus entirely on image datasets? Since the algorithm operates in an online fashion, it should, in theory, be able to handle very large datasets in chunks. There is a disconnect between the motivation of learning efficiently in the introduction section and the use of small datasets, where efficient methods are unnecessary, as full retraining only takes a couple of minutes. To properly transition from the presented motivation to the experiment section, at least one large dataset should be examined.

 W2. The evaluation metrics are not defined. What do the RA, UA, TA, JS, GAP initializations mean? Does MIA refer to Membership Inference Attack?

W3. Proper statistical testing is needed to confirm that the method indeed improves over the second best baseline in most cases. For example, looking at RA column in MNIST: the 0.5% improvement with std of 0.2% can be possibly due solely to noise. The 0.5% improvement in a test set of 10k images corresponds to a mere 50 images; this is not a large enough margin to provide convincing evidence that the method is actually advancing the SOTA.

 W4. Ablations: The ablation results in Table 2 are slightly concerning. In the text, the authors claim that "these experimental results indicate that the distribution shift loss and the forgetting data gradient contribute significantly to the unlearning process". However, looking at the tables, we often see that removing a component  actually improves performance metrics. Why is this? Does this not contradict the claim that all model components are useful, or at least provide conflicting evidence? There needs to be a more in-depth analysis and discussion of these results.

W5. Writing: The presentation of the paper could be improved significantly. In Table 1, it appears that white space partially covers some numbers.  Some figure captions need to provide much more information so that it is clear which results are being presented. The text in some of the figures is far too small, to the point of being illegible at normal magnification of the text. There are more grammatical errors and typos than would be expected in a submitted paper, especially with the availability of very effective grammar-checking tools.

**Questions:**

Q1. Eq. (1) and Line 111: The term $\mathcal{R} = 1/|F|\sum_{(x,y)\in F} d_{KL}(f(w), f(w^*))$ is confusing to me. If $w^*$ is defined as optimum $\mathcal{L}(D-L, w)$, wouldn't the KL term go to zero simply by minimizing the loss and then setting $w=\argmin \mathcal{L}(D-L, w)$?

Also, this seems at odds with Kurmanji et al., 2023. In their work they have a KL term on the subset of the data that they wish to forget weighted negatively (which makes sense because you would want to penalize similar outputs between the old and new models on that data subset) and positive KL penalty on the subset of the data they want to remember. This also makes sense because for that subset of data you want to prevent forgetting. Again, it would make things much easier to understand if there were a proper mathematical definition of “unlearning” in this paper.

Q2. What is the definition of forgetting for the authors? Please provide it clearly in the paper. Some papers define it as a statistical indistinguishability in model performance between unseen and "unlearned" data. In other works (Kurmanji et al., 2023 – see paper for full reference), "unlearning" relates to performing much worse on the unlearned data compared to the past while maintaining performance on the remaining data. This paper does not make clear the exact mathematical definition of unlearning that is being adopted. Kurmanji et al., 2023 has a good discussion on this issue.

Q3. I followed the proofs in the appendix carefully and they seem correct. I have a question regarding Theorem 2. Where does the inequality in Line 898 come from?

Q4. Eq. (5): Why is $w_t$ on the LHS? We are minimizing over $w_t$ on the RHS to obtain minimal risk. Also, $w_t^*$ should be the argmin not the min value of the loss.

Q5. Eq. (7): Second denominator: I assume $Q^{0}(y)$ should be $Q_{0}(y)$. Also, $Q(y|x)$ is defined but $Q(y)$ and $Q(x)$ are not. Can you please define them?

Q6. Eq. (12): Please define the norm in the main text to improve readability. It breaks the flow if the reader needs to go to appendix to determine which norm is used here.

---

> ### Author Response · Authors · 2024-11-24
> **Response to Reviewer 3cnv**
>
> Thank you for the reviews. Below we provide responses to the weaknesses raised as well as address the questions.
>
> **W1: While the theoretical grounding is excellent, the experiments are too small scale to be convincing and the statistical analysis is poor or missing...Why focus entirely on image datasets?**
>
> Thank you for your suggestion. Following your advice, we have extended our evaluation to the large-scale tinyImageNet dataset with ResNet18 to assess our proposed SAFE method against baselines for Remaining Accuracy (RA), Unlearning Accuracy (UA), Test Accuracy (TA), Membership Inference Attack (MIA), and computational time. For methods that do not require training data, we have highlighted in bold the performance values that are closest to the Retain method and exhibit significant differences, determined through a two-fold t-test at a 95% confidence level. For computational time, the fastest method has been bolded instead. In addition, We have applied a two-fold t-test at a 95% confidence level to all results presented in the paper and highlighted the significantly better values in bold.
>
> | Training Data Dependency                 | Method  | RA (%)       | UA (%)       | TA (%)       | MIA (%)      | Time (s) |
> |------------------------|---------|--------------|--------------|--------------|--------------|----------|
> | Training Data Required  | Retrain | 75.57 ± 0.51 | 42.71 ± 0.57 | 42.50 ± 0.54 | 29.69 ± 3.31 | 762.00   |
> |                        | Unroll  | 53.87 ± 2.00 | 48.62 ± 1.99 | 42.08 ± 1.42 | 24.20 ± 0.83 | 22.72    |
> |                        | CaMU    | 38.60 ± 2.14 | 36.17 ± 3.78 | 35.06 ± 1.98 | 39.07 ± 3.87 | 16.21    |
> | Training Data Not Required | LCODEC  | 9.34 ± 5.19  | 9.17 ± 5.21  | 6.84 ± 3.29  | 54.67 ± 3.62 | 13.04    |
> |                        | Descent-U | 27.50 ± 11.18| 27.48 ± 11.30| 24.63 ± 9.65 | 37.73 ± 6.75 | 20.70    |
> |                 | SAFE    | **48.69 ± 5.28** | **46.35 ± 4.05** | **36.28 ± 3.97** | **33.42 ± 2.17** | **7.30**     |
>
> Compared to LCODEC and Descent, which do not require access to training data during unlearning, SAFE significantly outperforms these methods in both effectiveness and efficiency. Importantly, SAFE achieves the fastest computation time at just 7.30 seconds, demonstrating its high efficiency. When compared to Unroll and CaMU (which require the original training data), SAFE's performance remains competitive while being much more efficient. Overall, SAFE strikes an excellent balance between accuracy and efficiency, particularly when training data is unavailable.
>
> We chose MNIST, Fashion-MNIST, and CIFAR-10 as these datasets are commonly used benchmarks in the literature for evaluating unlearning methods ([1], [2], [3]). They provide a standard baseline for comparing our results with prior work. Regarding NLP tasks, we have explored some and found that most NLP datasets rely on pretrained components, such as tokenizers, to encode information. Since unlearning a model trained from scratch is the primary focus of this work, the challenge of unlearning pretrained components is beyond the scope of the current study. In contrast, the widely used image datasets we evaluate have been shown to achieve high model performance without reliance on pretrained knowledge, making them well-suited for our focus. That said, we are still open to suggestions for NLP datasets that do not depend heavily on pretrained components and would be happy to explore them.
>
>
>
> **W2: The evaluation metrics are not defined. What do the RA, UA, TA, JS, GAP initializations mean? Does MIA refer to Membership Inference Attack?**
> Thank you for pointing this out. The initilizations and metrics mean
> - RA (Remaining Accuracy): The prediction accuracy of the model on the remaining data after unlearning.
> - UA (Unlearning Accuracy): The prediction accuracy of the model on the forgetting data after unlearning.
> - TA (Test Accuracy): The prediction accuracy of the model on the overall test data after unlearning.
> - JS (Jensen–Shannon Divergence): Measures the divergence between the outputs of the model after unlearning and the outputs of the model retrained on the remaining data. Lower JS indicates smaller differences between the two outputs, signifying better unlearning performance.
> - MIA (Membership Inference Attack): Refers to the model’s vulnerability to membership inference attacks.
>
> We have updated them in the paper (Lines 378 to 388).

---

> ### Author Response · Authors · 2024-11-24
> **Response to Reviewer 3cnv**
>
> **W3: Proper statistical testing is needed to confirm that the method indeed improves over the second-best baseline in most cases.**
>
> Thank you for highlighting this point. Following the suggestion, we conducted statistical testing on all the experimental results presented in Table 1. Methods that are significantly better than others are bolded based on a two-tailed t-test with a 90% confidence level. For instance, the p-values for RA, UA, and TA on the CIFAR-10 dataset are 0.0020, 0.0380, and 0.0008, respectively. These results demonstrate that the proposed method performs significantly better than the other methods.
>
> Additionally, the p-values for RA, TA, and JS on the MNIST dataset are 0.0087, 0.0130, and 0.0919, respectively, and the p-values for RA, UA, TA, and JS on the FMNIST dataset are 0.0056, 0.0008, 0.0075, and 0.0342, respectively. These findings further confirm that the proposed method outperforms Unroll with at least 90% confidence.
>
> To address the concern of 0.5% accuracy improvement and provide a more robust evaluation of the method's performance, we employ the Jensen–Shannon Divergence (JS) as an additional metric. Unlike accuracy, which evaluates only the correctness of discrete labels, JS captures the differences in the real-valued output distributions of the model. This approach reveals a substantially greater improvement (for example, SAFE can achieve less than 3.5 Remain JS while other baselines can only achieve 4.0 or higher Remain JS), showing that our proposed SAFE method is advancing SOTA.
>
> **W4: Ablations: The ablation results in Table 2 are slightly concerning. We often see that removing a component actually improves performance metrics. Why is this? Does this not contradict the claim that all model components are useful, or at least provide conflicting evidence?**
>
> Unlearning is inherently a dual-objective challenge: it requires maintaining the performance on both the retained data and test data while effectively unlearning the forgetting data. In our proposed method, each component plays a distinct role, contributing to specific aspects of the problem. Collectively, however, the proposed SAFE method achieves a balance between these dual objectives. In the following discussion, we elaborate on the role of each component and how the ablation study reflects their individual contributions.
>
> - DSR, which stands for Distribution Shift Risk as presented in Table 2, is specifically designed for unlearning on the forgetting data. Without the DSR component, the JS and UA metrics—used to evaluate the performance on the forgetting data—are worse than those of the SAFE method. This demonstrates DSR's effectiveness in reducing performance on the forgetting data, aligning with its intended functionality.
> - TR, representing the initial training risk, is essential for maintaining the performance of both the remaining data and the test data. The ablation study results reveal a more significant performance reduction without TR compared to the SAFE algorithm, demonstrating its role.
> - FR, which stands for forgetting data risk, requires further evaluation. The results in Table 2 are presented as averages across all iterations in the streaming process. In Eq. 11, the FR component is calculated based on the forgetting data, which is relatively small at the beginning. This smaller size may compromise the reliability of FR's evaluation during initial iterations, leading to a less reliable average value in Table 2. To address this, we conducted an additional ablation study focusing on the final iteration, where sufficient forgetting data has accumulated to provide a more reliable estimation. The results, shown in the following tables using CIFAR10 data with the Logistic Regression (LR) and ResNet models, indicate that removing the FR component results in a higher UA—reflecting increased accuracy and reduced unlearning on the forgetting data. These additional results have been included in the appendix.
>
> | CIFAR10- LR  | RA (%)       | UA (%)       | TA (%)       | JS (%)      |
> |---------|--------------|--------------|--------------|--------------|
> | Retrain | 85.30 ± 0.72 | 82.68 ± 2.57 | 84.48 ± 0.86 | 0.00 ± 0.00 | 762.00   |
> | w/o DSR | 85.92 ± 0.13| 84.80 ± 2.65| 85.05 ± 0.25 | 3.02 ± 0.23 |
> | w/o FR | 85.82 ± 0.85| 83.59 ± 2.83| 84.88 ± 0.24 | 1.69 ± 0.14 |
> | w/o TR | 85.40 ± 0.35| 82.20 ± 1.84| 84.55 ± 0.06 | 1.23 ± 0.14 |
> | SAFE    | 85.77 ± 0.23 | 83.07 ± 2.83 | 84.83 ± 0.32 | 1.41 ± 0.07 |
>
> | CIFAR10 - ResNet | RA (%)       | UA (%)       | TA (%)       | JS (%)      |
> |---------|--------------|--------------|--------------|--------------|
> | Retrain | 97.18 ± 0.23 | 91.12 ± 0.56 | 90.65 ± 0.41 | 0.00 ± 0.00 | 762.00   |
> | w/o DSR | 97.35 ± 0.18| 97.46 ± 0.27| 90.83 ± 0.25 | 4.20 ± 0.26 |
> | w/o FR | 93.35 ± 1.03| 93.00 ± 1.70| 86.86 ± 1.08 | 6.05 ± 0.50 |
> | w/o TR | 90.92 ± 1.53| 88.92 ± 2.04| 84.68 ± 1.38 | 7.32 ± 0.77 |
> | SAFE    | 93.70 ± 1.29 | 92.34 ± 1.92 | 97.19 ± 1.28 | 5.85 ± 0.64 |

---

> ### Author Response · Authors · 2024-11-24
> **Response to Reviewer 3cnv**
>
> **Q1: Eq. (1) and Line 111: The term** $\mathcal{R}(F, w) = {1}/{|F|}\sum_{(\mathbf{x},y)\in F} d_{\text{KL}}(f(\mathbf{x};w), f(\mathbf{x};w^*))$ **is confusing to me. If** $w^*$ **is defined as optimum** $\mathcal{L}(D-L,w)$**, wouldn't the KL term go to zero simply by minimizing the loss and then setting** $w^* = \arg\min\mathcal{L}(D-L,w)$? **Also, this seems at odds with Kurmanji et al., 2023. In their work they have a KL term on the subset of the data that they wish to forget weighted negatively.**
>
> We posit that $w^* = \arg\min \mathcal{L}(D-L, w)$ is also an optimizer for our proposed Eq. (1). Theoretically, optimizing Eq. (1) could lead to $w^* = \arg\min \mathcal{L}(D-L, w)$, suggesting that the KL divergence term may not be necessary.
>
> However, as one of the key objectives is to achieve unlearning specifically on the forgetting data, we observe that optimizing $w^* = \arg\min \mathcal{L}(D-L, w)$ alone in a streaming setting, as shown by the w/o DSR line in Table 2, does not achieve unlearning accuracy (UA) as effectively as true retraining. This discrepancy can be attributed to the lack of constraints on the performance for the forgetting data. It is also worth noting that setting $\lambda = 0$ reduces the problem to unlearning via $\min \mathcal{L}(D-L, w)$. In this work, we propose a more general framework that allows flexibility to specifically consider the forgetting data in a streaming setting—an approach that has proven effective in our experiments considering various metrics.
>
> While our objective aims to fully align with the retrained model—both on forgetting data and remaining data—Kurmanji et al. (2023) explicitly design their objective to produce poor predictions on the forgetting data, leading to large errors for those data points. As stated in your raised Q2, their method adopts a fundamentally different unlearning objective compared to retraining, which we have not yet explored in a streaming setting in this paper. The negative term in the objective proposed by Kurmanji et al. (2023) may introduce challenges in optimization under a streaming setting. We appreciate the comment highlighting this alternative approach and plan to explore it in future work.
>
>
>
> **Q2: What is the definition of forgetting for the authors? Please provide it clearly in the paper.**
>
> Thank you for raising this point. In this work, we define unlearning as the model that, after unlearning, should perform similarly to the model retrained solely on remaining data. The two models should show close performance on both remaining training data and other unseen data, such as forgetting or test data. Our definition is consistent with the ones in prior works ([2], [4], [5]), which describe general machine unlearning as "Unlearning guarantees that training on a point and unlearning it afterwards will produce the same distribution of models as not training on the point at all." This also emphasizes the importance of achieving performance similar to a retrained model on both the remaining and forgetting data predictions.
>
> Based on the definition, the straightforward objective for unlearning is $w^* = \arg\min \mathcal{L}(D-L, w)$, which can ensure the post-unlearning model is close to the retrained model on the remaining data through retraining. But to effectively ensure that the forgetting data can also perform similarly to the retrained model, the objective can be formulated as:
> $$
> w^* = \arg\min \mathcal{L}(D-L, w) + \lambda \mathcal{R}(F, w)
> $$
> where the regularizer $\mathcal{R}(F, w)$ is defined as:
> $$
> \mathcal{R}(F, w) = \frac{1}{|F|} \sum_{(\mathbf{x}, y) \in F} d_{\text{KL}}(f(\mathbf{x}; w), f(\mathbf{x}; w^*)),
> $$
> and imposes restrictions on the forgetting data $F$ by encouraging divergence from the original model's outputs.
>
> By optimizing the regularizer, we align the post-unlearning model's outputs on the forgetting data with the retrained model, ensuring forgetting is enforced.
>
>
> **Q3: I followed the proofs in the appendix carefully and they seem correct. I have a question regarding Theorem 2. Where does the inequality in Line 898 come from?**
>
> It comes from the upper-bounded gradient assumption in Theorem 1, where we assume $||\nabla R_{t}(w)||\le U$. Following this assumption, then gradient from $w$ to $w^*$, that is $\frac{|R_{t}(w) - R_{t}(w^*)|}{|w - w^*|}$ will also be less or equal than $U$. We have made it clear in the proof.
>
> **Q4: Eq. (5): Why is $w_t$ on the LHS?**
>
> It should be $w$ on the LHS and $\arg\min$ in the RHS. Thank you for raising this.

---

> > ### Author Response · Authors · 2024-11-24
> > **Response to Reviewer 3cnv**
> >
> > **Q5: Eq. (7): Second denominator: I assume $Q^0(y)$ should be $Q_0(y)$. Also, $Q(y|x)$ is defined but $Q(y)$ and $Q(x)$ are not. Can you please define them?**
> >
> > $Q^0(y)$ should be $Q_0(y)$. $Q(x)$ denotes the feature marginal distributions on feature $x$ and $Q(y)$ stands for the proportions of the data belonging to class $y$. We have added the definition in the revised versions.
> >
> >
> >
> > **Q6: Eq. (12): Please define the norm in the main text to improve readability. It breaks the flow if the reader needs to go to the appendix to determine which norm is used here.**
> >
> > We have moved the norm definition from the appendix to the main text. Thank you for the suggestion.
> >
> >
> >
> > [1] Tarun, et al. "Fast yet effective machine unlearning." TNNLS 2023.
> >
> > [2] Bourtoule, et al. "Machine unlearning." SP, 2021.
> >
> > [3] Chundawat, et al. "Zero-shot machine unlearning." IEEE Transactions on Information Forensics and Security, 2023.
> >
> > [4] Sekhari, et al. "Remember what you want to forget: Algorithms for machine unlearning." NeurIPS, 2021.
> >
> > [5] Fan, et al. "SalUn: Empowering Machine Unlearning via Gradient-Based Weight Saliency in Both Image Classification and Generation." ICLR, 2024.

---

> > > ### Comment · Reviewer_3cnv · 2024-11-25
> > > **Acknowledgement of the response**
> > >
> > > I thank the authors for the detailed response. While many of my concerns have been addressed, the responses to Q1 and Q2 remain problematic.
> > >
> > > Concerning Q1, the response states that it "suggest[s] that the KL divergence term may not be necessary." Based on the arguments in the response, it goes beyond a suggestion, and seems to directly imply this. This undermines the initial formulation of the problem. Why include an extra term in an optimization problem if it is known that it is irrelevant for the optimal solution? While this formulation may indeed lead to better empirical performance in a streaming setting, as a stand-alone optimization problem, it doesn't make sense. The paper doesn't introduce the addition of the KL term as a mechanism to improve stability/performnace in a streaming setting. It is presented as though it is a meaningful addition to the static optimization problem.
> > >
> > > Concerning Q2, the provided definition of unlearning is still not mathematical or rigorous - "as the model that, after unlearning, should perform similarly to the model retrained solely on remaining data. The two models should show close performance on both remaining training data and other unseen data, such as forgetting or test data." What does "similarly" mean? What does "close performance" mean?  What are the mathematical definitions of these expressions?
> > >
> > > The paper by Sekhari et al. [4] has a precise definition of unlearning (Definition 2). There are other precise definitions of forgetting/unlearning in the literature: Definiton 3.1 in [R1]; Section 2 – certified removal definition in [RR2].
> > >
> > > Neither the response nor the revised paper provides a precise definition of unlearning. There is no concrete support for the statement in the response that “Our definition is consistent with the ones in prior works” – especially with regard to [4], which does have a mathematical definition. If there is a claim of consistency, then it is necessary to (a) provide a formal definition of unlearning; and (b) establish that it is (in some sense) the same as the definition in [4].
> > >
> > > Without a mathematical definition of the unlearning problem, it is not clear whether the optimization problem (1) actually aligns with the desired task. As such, as argued by one of the other reviewers, it is a heuristic approach, which, while yielding impressive empirical performance, does not have a solid theoretical foundation or motivation.
> > >
> > > [RR1] Ginart et al., “Making AI Forget You: Data Deletion in Machine Learning,” NeurIPS 2019.
> > >
> > > [RR2] Guo et al., “Certified Data Removal from Machine Learning Models,” ICM 2020.

---

> ### Author Response · Authors · 2024-12-02
>
> Thank you for your continued feedback.
>
> **Clarifications on "Similar" or "Close Performance**
>
> Firstly, we would like to clarify the statements regarding "similar" or "close performance." By this, we mean that the outputs of the unlearned model are comparable to the outputs of the retrained model on the same data [FR1].
>
> **Response to Q1: KL Divergence Term and Forgetting Data Performance**
>
> Regarding the concerns about Q1, our intention in adding the KL divergence term is to impose restrictions on the performance for the forgetting data. While it may not directly influence the optimization results of: $w^* = \arg\min \mathcal{L}(D-L, w),$ the additional regularizer term further ensures that the divergence between the predictions on the forgetting data and those of the retrained model is minimized. Without this regularizer term, although the performance of the model on the remaining data after unlearning may approach that of the retrained model, the performance on the forgetting data could significantly deviate from that of the retrained model.
>
> For instance, in the case of an overfitting model where the loss on the remaining data ($\mathcal{L}(D-L, w)$) has been minimized during initial training, the gradients on $\mathcal{L}(D-L, w)$ might be negligible, which would result in unlearning being insufficiently achieved through $\mathcal{L}(D-L, w)$ alone. In contrast, the KL term plays a critical role, particularly in stream settings, by ensuring that the performance on forgetting data aligns closely with the retrained model through all rounds of unlearning requests.
>
> **Connection with Certified Unlearning**
>
> Regarding the relationship between our work and the mathematical definition of certified unlearning, the bounded gradient assumption in Theorem 2 of our work satisfies the requirements of **Theorem 2 in [FR2]**. Consequently, it can be proven that our proposed unlearning method constitutes an $(\epsilon, \delta)$-certified removal.
>
> We appreciate your suggestion to provide a formal proof for this connection. In the revised paper, we will include the relevant proof to establish this connection clearly.
>
>
> [FR1] Fan, et al. "SalUn: Empowering Machine Unlearning via Gradient-Based Weight Saliency in Both Image Classification and Generation." ICLR, 2024.
>
> [FR2] Guo, et al. “Certified Data Removal from Machine Learning Models.” ICML, 2020.

---

> > ### Comment · Reviewer_3cnv · 2024-12-03
> > **Thank you for the clarification**
> >
> > Thank you for the response.
> >
> > The inclusion of a clear, mathematical definition of unlearning would definitely make the paper considerably stronger. This would first and foremost make the problem statement clear. The proof of a connection to certified unlearning would then significantly add to the paper and provide a strong motivation for the proposed technique. It would also provide an avenue for explaining exactly why the KL divergence term is included (in a principled way). While your response provides some explanation, I don't think the inclusion of this term in the static optimization problem is the best way to begin the paper.
> >
> > It would be best to avoid words/phrases like "similarly", "close performance", or "comparable" in a mathematical definition. None of these are quantifiable, so it becomes difficult to determine whether unlearning has been achieved or not.
> >
> > Overall, the paper has some nice ideas and some promising results. The proposed changes constitute relatively major revisions, and I think another review would be necessary - certainly a claimed theoretical result, even if the proof is relatively straightforward, can't really be accepted without an inspection of the proof. With this in mind, I will retain the score I originally assigned for now, but because I have a favourable view of the contributions of the paper, I will keep an open mind during the discussion with the other reviewers.

---

### Official Review · Reviewer_zsVD · 2024-11-02

**Soundness:** 2
**Presentation:** 2
**Contribution:** 2
**Rating:** 5
**Confidence:** 3

**Summary:**

This paper focuses on machine unlearning, in particular in the settings where requests for data removal arise in a streaming manner. To address the new challenges of performance maintenance, efficiency and data access brought about by this new streaming setting, this paper introduced a new online unlearning paradigm. Specifically, a new algorithm named SAFE was proposed. The performance of SAFE was analyzed with a regret upper bound guarantee, and numerically evaluated and compared to baseline methods.

**Strengths:**

- This paper considers a practical setting for machine unlearning, i.e., the streaming setting where new data may continue arrive or be generated during the unlearning process. This new settings bring many new challenges compared to the widely-studied static/batched settings. This paper addressed these new challenges via proposing a new SAFE algorithm.
- The performance of SAFE algorithm was (a) analyzed with a regret upper bound guarantee; and (b) numerically evaluated using several datasets and compared to baseline methods.

**Weaknesses:**

- The performance analysis of SAFE is relatively weak or may be very straightforward given the existing literature. For instance, the bound on the error of the t-th rounds is simply bounded by $O(\sqrt{T})$, which is straightforward and should not be a tight bound.
- As noted by the authors, the accumulated regret is also not tight and compared to existing results.
- Despite that MNIST, FASHION, and CIFAR-10 have been widely used, it may be more interesting to consider more complex datasets such as (Tiny)-ImageNet. In addition, all these datasets focus on the image classification task, it may be also interesting to consider datasets for other tasks, such as NLP tasks.

**Questions:**

- Can you elaborate the tightness of your regret bounds? Also what are the coefficients associated in the bounds with respect to $T$? Should the bound be written as $\tilde{\mathcal{O}}(\sqrt{T})$?
- How do you know the value of $V_T$ in advance? which depends on all the $T$ rounds and the data arrive in a streaming manner? Likewise for the value of $\Delta_T$. In addition, are their values constant and independent of $T$?
- Can your algorithm operate without knowing the values of $V_T$ and $\Delta_T$ in advance?

**Details Of Ethics Concerns:**

N/A.

---

> ### Author Response · Authors · 2024-11-24
> **Response to Reviewer zsVD**
>
> Thank you for the reviews. Below we provide responses to the weaknesses raised as well as address the questions.
>
> **W1 The performance analysis of SAFE is relatively weak or may be very straightforward, given the existing literature. For instance, the bound on the error of the t-th rounds is simply bounded by $O(\sqrt{T})$, which is straightforward and should not be a tight bound;**
>
> We appreciate the reviewer’s feedback on SAFE's performance analysis. However, we respectfully disagree with the assessment that "*the performance analysis of SAFE is relatively weak or may be very straightforward given the existing literature*" and that "*$O(\sqrt{T})$, which is straightforward and should not be a tight bound*."
>
> To the best of our knowledge, this is the first work to provide a regret bound for the classification unlearning problem. Deriving these bounds required us to innovatively reformulate each round's unlearning process as a distribution shift problem and conduct the corresponding analysis about the potential range under different unlearning tasks like class-wise or instance-wise unlearning. This approach is not trivial and represents one of the key technical contributions of our work.
>
> In comparison to existing unlearning literature, [1] focuses on the batch unlearning setting and provides an error bound that scales quadratically with the size of the unlearning set. When extended to streaming unlearning, the unlearning set size typically grows linearly with $T$, resulting in a regret bound of $O(T^2)$. In contrast, our regret bound achieves a significant improvement, ranging from $O(T^{3/2})$ to $O(\sqrt{T})$ (see more details in our response to Q1). This improvement demonstrates that our bound is tighter than those in existing literature.
>
>
> **W2 As noted by the authors, the accumulated regret is also not tight and compared to existing results.**
>
> While we have discussed some results from online learning in our paper and compared their bounds to our own, we would like to clarify that we did not characterize our regret bound as a weakness. On the contrary, we view it as a key contribution of our work, offering a significant improvement over a straightforward extension of existing results, as detailed in our response to W1.
>
> While it is true that some online learning works have achieved bounds with lower dependence on $T$, these results cannot be directly extended to the unlearning problem without significant adaptation. The unique challenges and objectives of the unlearning problem distinguish it from standard online learning. Firstly, in machine unlearning, the initial model already has comprehensive knowledge about all the training data, including the data to be forgotten.Secondly, the availability of training data in online learning and online unlearning differs. Thirdly, from a practical standpoint, our online unlearning method, SAFE, does not assume any convexity or pseudo-convexity for the training loss. The detailed discussion has been provided in Line 155-161
>
>
> Our discussion of online learning results in the paper is intended to provide context and insights for readers who may wish to explore these connections further, rather than as a direct comparison.
>
> We hope this response clarifies the novelty and significance of our contribution and provides additional context for our discussion of related literature in online learning. We also remain open to suggestions for how the connections between unlearning and online learning could be better framed in future work.

---

> > ### Author Response · Authors · 2024-11-24
> > **Response to Reviewer zsVD**
> >
> > **W3 Despite that MNIST, FASHION, and CIFAR-10 have been widely used, it may be more interesting to consider more complex datasets such as (Tiny)-ImageNet. In addition, all these datasets focus on the image classification task, it may be also interesting to consider datasets for other tasks, such as NLP tasks.**
> >
> > Thank you for your suggestion. Following your advice, we have extended our evaluation to include tinyImageNet with ResNet18 to assess our proposed SAFE method against baselines for Remaining Accuracy (RA), Unlearning Accuracy (UA), Test Accuracy (TA), Membership Inference Attack (MIA), and computational time. For all methods that do not require training data, we have highlighted in bold the performance closest to the Retrain method, except for computational time, where we have highlighted the quickest one.
> >
> > | Training Data Dependency                 | Method  | RA (%)       | UA (%)       | TA (%)       | MIA (%)      | Time (s) |
> > |------------------------|---------|--------------|--------------|--------------|--------------|----------|
> > | Training Data Required  | Retrain | 75.57 ± 0.51 | 42.71 ± 0.57 | 42.50 ± 0.54 | 29.69 ± 3.31 | 762.00   |
> > |                        | Unroll  | 53.87 ± 2.00 | 48.62 ± 1.99 | 42.08 ± 1.42 | 24.20 ± 0.83 | 22.72    |
> > |                        | CaMU    | 38.60 ± 2.14 | 36.17 ± 3.78 | 35.06 ± 1.98 | 39.07 ± 3.87 | 16.21    |
> > | Training Data Not Required | LCODEC  | 9.34 ± 5.19  | 9.17 ± 5.21  | 6.84 ± 3.29  | 54.67 ± 3.62 | 13.04    |
> > |                        | Descent | 27.50 ± 11.18| 27.48 ± 11.30| 24.63 ± 9.65 | 37.73 ± 6.75 | 20.70    |
> > |                 | SAFE    | **48.69 ± 5.28** | **46.35 ± 4.05** | **36.28 ± 3.97** | **33.42 ± 2.17** | **7.30**     |
> >
> > Compared to LCODEC and Descent, which do not require access to training data during unlearning, SAFE significantly outperforms these methods in both effectiveness and efficiency. Importantly, SAFE achieves the fastest computation time at just 7.30 seconds, demonstrating its high efficiency. When compared to Unroll and CaMU (which require the original training data), SAFE's performance remains competitive while being much more efficient. Overall, SAFE strikes an excellent balance between accuracy and efficiency, particularly when training data is unavailable.
> >
> > Regarding NLP tasks, we have explored some and found that most NLP datasets rely on pretrained components, such as tokenizers, to encode information. Since unlearning a model trained from scratch is the primary focus of this work, the challenge of unlearning pretrained components is beyond the scope of the current study. In contrast, the widely used image datasets we evaluate have been shown to achieve high model performance without reliance on pretrained knowledge, making them well-suited for our focus. That said, we are still open to suggestions for NLP datasets that do not depend heavily on pretrained components and would be happy to explore them.

---

> > > ### Author Response · Authors · 2024-11-24
> > > **Response to Reviewer zsVD**
> > >
> > > **Q1 Can you elaborate the tightness of your regret bounds? Also what are the coefficients associated in the bounds with respect to $T$? Should the bound be written as $O(\sqrt{T})$? (should be answered together with Q2)**
> > > and
> > > **Q2 How do you know the value of $V_T$ in advance? which depends on all the $T$ rounds and the data arrive in a streaming manner? Likewise for the value of $\Delta_T$. In addition, are their values constant and independent of $T$?**
> > >
> > > $V_T = \sum_{t=0}^T \|w_t^* - w_{t+1}^*\|$ represents the cumulative total variation in the optimal parameters $w_t^*$ over $T$ unlearning rounds. $V_T$ cannot be known in advance. Following [x1], we analyze the behavior of $V_T$ as follows
> > > - Constant $O(1)$: For tasks with minimal parameter variation over time (e.g., static or nearly stationary problems), $V_T$ remains constant.
> > > - Sublinear to $T$ ($O(T^\gamma)$, $\gamma<1$): For tasks where gradients grow smoothly as $T$ increases (e.g., problems with diminishing changes in the loss landscape over time), $V_T$ grows sublinearly.
> > > - Linear to $T$ ($O(T)$): For tasks where parameter variation remains stable and unlearning requests introduce consistent changes, $V_T$ scales linearly with $T$.
> > >
> > > Based on the above analysis, the $O(V_t\sqrt{T})$ ranges from $O(\sqrt{T})$ to $O(T^{3/2})$, depending on the specific properties of the task and unlearning dynamics.
> > >
> > > $\Delta_T$ is the cumulative total divergence between the remaining and forgetting data distributions across $T$ unlearning rounds. It can range from $O(1)$ to $O(T)$, depending on the divergence between the remaining and forgetting data distributions.
> > > - Constant $O(1)$ when remaining and forgetting data distributions are close, for example, when forgetting data are uniformly randomly sampled from the training data.
> > > - Linear to $T$ ($O(T)$) when the remaining and forgetting data distributions are totally disjoint, for example, in the class-wise unlearning, where the remaining and forgetting data belong to different classes.
> > >
> > > Overall, the regret bound can range from $O(\sqrt{T})$ to $O(T^{3/2})$. Even in the worst-case scenario, it is still tighter than $O(T^2)$, which would result from a straightforward extension of results in [x2] (as detailed in our response to W1).
> > >
> > > **Q3 Can your algorithm operate without knowing the values of $V_T$ and $\Delta_T$ in advance?**
> > >
> > > Yes, it can. These values reflect the inherent properties of the unlearning problem itself, such as the data distribution discrepancy between the forgetting and remaining data, as well as the relative parameter differences across rounds in the streaming process. They are primarily used to establish the theoretical guarantees of the algorithm but are not required as inputs for the algorithm's operation.
> > >
> > > [1] Sekhari, et al. "Remember what you want to forget: Algorithms for machine unlearning." NeurIPS, 2021.

---

> > > > ### Comment · Reviewer_zsVD · 2024-11-26
> > > > **Thank you for the rebuttal**
> > > >
> > > > Thank you for your rebuttal, which addressed some of my concerns. (Not for machine unlearning) there are many recent works studying the non-stationary settings for multi-armed bandits (or contextual bandits, linear bandits) and markov decision process (MDP) and constrained (MDP), it is well-known open challenges for both the design of online algorithms and the theoretical proofs when the variations are unknown in advance. Although I am not pretty sure, I am a bit unconvinced about the argument that your algorithm can be easily extended to the case without knowing the values of $V_T$ and $\Delta_T$ in advance, and they are purely used for theoretical analysis.
> > > >
> > > > After reading comments from other reviewers, I will keep the current rating.

---

> > > > > ### Author Response · Authors · 2024-12-02
> > > > >
> > > > > Thank you for your feedback. Here, we want to clarify that in the optimization of our methods, we only need to get the upper bound of parameters ($W$ in Theorem 2) and the total unlearning rounds $T$ to calculate the learning rate $\gamma$. $W$ can be obtained from the well-trained model, and $T$ depends on the total length of stream forgetting request. We do not require the $V_T$ and $\Delta_T$ to be known to conduct the online optimizations.
> > > > >
> > > > > Thank you again for your valuable time and insightful comments!

---

### Official Review · Reviewer_nwmJ · 2024-11-03

**Soundness:** 3
**Presentation:** 3
**Contribution:** 2
**Rating:** 6
**Confidence:** 3

**Summary:**

In this paper, the authors consider a online machine unlearning problem, where requests for data remove arise in a streaming manner rather than in a single batch. This causes new challenges, and this paper solves this problem by viewing unlearning as a distribution shift problem. They estimate the altered distribution and propose a online unlearning method without having access to the original training data. Theoretical analysis is also conducted without the strong restriction of convex loss function. Experiments are conducted using 4 different datasets, confirming the advantage of the proposed method.

**Strengths:**

1. This paper is well written and easy to follow. The presentation and the organization of the paper is good.

2. This paper tackles an important problem, which is to conduct unlearning in a streaming manner. This problem has not been well studied in the unlearning literature.

3. The idea of viewing the unlearning as distribution shift problem has not been well explored before. The approach is relatively novel compared to existing works.

4. The paper also provides theoretical insights into the proposed online unlearning methodology.

**Weaknesses:**

1. Technical contributions: In Line 65, the authors mention that only two studies (Zhao et al., 2024; Li et al., 2021) have directly tackled the problem of stream data forgetting. However, it is not clear what are the new technical novelties of this work compared to previous works. It would be great if the authors can better illustrate their technical contributions and advantages in the revised paper.

2. Baselines: During experiments, it is not clear why the above two works (Zhao et al., 2024; Li et al., 2021) are not considered as baselines. It would be good to make a comparison with these closely related works.

3. Datasets: The datasets considered in this paper is relatively small. I recommend the authors to consider ImageNet related datasets, at least miniImageNet or tinyImageNet.

**Questions:**

My questions can be found in the weakness above.

---

> ### Author Response · Authors · 2024-11-24
> **Response to Reviewer nwmJ**
>
> Thank you for the reviews. Below, we provide responses to the three weaknesses raised.
>
> **W1. New technical novelties of this work compared to previous works (Zhao et al., 2024; Li et al., 2021).**
>
> The main technical contribution of our work is that it is the first to address the problem of streaming unlearning for classification models without requiring access to the original training data during the unlearning process. Moreover, our approach does not restrict the classification model architecture, making it broadly applicable. In comparison, Zhao et al. (2024) focus on object detection tasks and rely on a transformer-based structure, while Li et al. (2021) address regression tasks using linear models. Both of these methods require repeated access to the original training data in each round of the streaming phase, making them far less practical in scenarios where diverse model architectures are used and accessing data repeatedly is either impractical or impossible due to privacy concerns.
>
> In addition, we present a regret risk bound for our approach to the classification problem. Specifically, we show that the accumulated error can be bounded by $O(V_T\sqrt{T}+ \Delta_T)$. In contrast, simply extending existing results from prior work [1] would lead to a significantly less favourable regret bound of $O(T^2)$. While Zhao et al. (2024) do not provide any theoretical results, Li et al. (2021) offer a bound for the linear regression problem, which does not apply to the current classification problem and non-linear models.
>
> [1] Sekhari, et al. "Remember what you want to forget: Algorithms for machine unlearning." NeurIPS, 2021.
>
> **W2. Why the above two works (Zhao et al., 2024; Li et al., 2021) are not considered as baselines.**
>
> The two works (Zhao et al., 2024; Li et al., 2021) were not considered as baselines in our experiments because their focus, methodologies, and requirements differ fundamentally from our approach. Our work is designed specifically for streaming unlearning in classification tasks without accessing the original training data, whereas Zhao et al. and Li et al. require access to training data during unlearning and focus on different types of learning problems (object detection and regression, respectively).
>
>
> **W3. Consider ImageNet related datasets, at least miniImageNet or tinyImageNet.**
>
> Thank you for your suggestion regarding the dataset. Following your advice, we have extended our evaluation to include tinyImageNet with ResNet18 to assess our proposed SAFE method against baselines for Remaining Accuracy (RA), Unlearning Accuracy (UA), Test Accuracy (TA), Membership Inference Attack (MIA), and computational time. For all methods that do not require training data, we have highlighted in bold the performance closest to the retrained method, except for computational time, where we have highlighted the quickest one.
>
> | Training Data Dependency                 | Method  | RA (%)       | UA (%)       | TA (%)       | MIA (%)      | Time (s) |
> |------------------------|---------|--------------|--------------|--------------|--------------|----------|
> | Training Data Required  | Retrain | 75.57 ± 0.51 | 42.71 ± 0.57 | 42.50 ± 0.54 | 29.69 ± 3.31 | 762.00   |
> |                        | Unroll  | 53.87 ± 2.00 | 48.62 ± 1.99 | 42.08 ± 1.42 | 24.20 ± 0.83 | 22.72    |
> |                        | CaMU    | 38.60 ± 2.14 | 36.17 ± 3.78 | 35.06 ± 1.98 | 39.07 ± 3.87 | 16.21    |
> | Training Data Not Required | LCODEC  | 9.34 ± 5.19  | 9.17 ± 5.21  | 6.84 ± 3.29  | 54.67 ± 3.62 | 13.04    |
> |                        | Descent | 27.50 ± 11.18| 27.48 ± 11.30| 24.63 ± 9.65 | 37.73 ± 6.75 | 20.70    |
> |                 | SAFE    | **48.69 ± 5.28** | **46.35 ± 4.05** | **36.28 ± 3.97** | **33.42 ± 2.17** | **7.30**     |
>
> Compared to LCODEC and Descent, which do not require access to training data during unlearning, SAFE significantly outperforms these methods in both effectiveness and efficiency. Importantly, SAFE achieves the fastest computation time at just 7.30 seconds, demonstrating its high efficiency. When compared to Unroll and CaMU (which require the original training data), SAFE's performance remains competitive while being much more efficient. Overall, SAFE strikes an excellent balance between accuracy and efficiency, particularly when training data is unavailable.

---

> > ### Comment · Reviewer_nwmJ · 2024-11-25
> >
> > Thanks for the response. After reading the response and other reviewers' comments, I have decided to keep my original score 6, marginally above the acceptance threshold.

---

> > > ### Author Response · Authors · 2024-12-02
> > >
> > > Thank you again for your valuable time and insightful comments and suggestions!

---

### Author Response · Authors · 2024-11-24
**Global response**

We thank the reviewers for their precious time and insightful comments that improved the exposition and readability of our paper. We have made the following changes in our revision accordingly:

1. We add more detailed descriptions of the evaluation metrics in Section 4.1.
2. We add statements for the significance of the experiment results in Table 1 based on the comments of Reviewer 3cnv.
3. We move MIA results and analysis to Section 4.2 to address the weakness of Reviewer 3ki7. We move the complete results table from Appendix to Section 4.2.
4. We add extra experiments on TinyImagenet and present the results in Appendix B.9.
5. We fix the column width in the tables and add more descriptions in the captions of figures based on the comments of Reviewer 3cnv.
6. We fix some typos in the technical part.

---

### Meta-Review · Area_Chair_Shni · 2024-12-11

**Metareview:**

This paper addresses the problem of machine unlearning in a streaming setting, where data removal requests occur sequentially over time. The authors propose a novel unlearning algorithm, SAFE, which treats unlearning as a distribution shift problem and does not require access to the original training data. The paper provides a theoretical regret bound for the algorithm and validates it through empirical experiments on several datasets.

Reviewers generally acknowledged the novelty of applying a streaming unlearning approach and appreciated the theoretical analysis and empirical validation. However, the lack of a rigorous definition of unlearning and the superlinear regret bound were major concerns. The proposed regret bound was seen as suboptimal, and reviewers questioned its practical relevance.

**Additional Comments On Reviewer Discussion:**

After discussions, the authors have addressed some concerns, but the following issues remain:

The authors still have not provided a clear mathematical definition of unlearning. The claimed theoretical result lacks a formal proof. Additionally, the empirical evidence is not convincing enough to support the paper's claims.

---

### Decision · Program_Chairs · 2025-01-22

Reject